



# In depth characterization of diazotroph activity across the
# Western Tropical South Pacific hot spot of $N_2$ fixation
Sophie Bonnet[1,2], Mathieu Caffin[1], Hugo Berthelot[1], Olivier Grosso[1,] Mar Benavides[2,3], Sandra
Helias-Nunige[2], Cécile Guieu[4,5], Marcus Stenegren[6], Rachel A Foster[6]
[1]IRD, Aix Marseille Université, CNRS/INSU, Université de Toulon, Mediterranean Institute of Oceanography (MIO)
UM 110, 13288, Marseille-Nouméa, France-New Caledonia
[2]Mediterranean Institute of Oceanography (MIO) – IRD/CNRS/Aix-Marseille University IRD Nouméa, 101
Promenade R. Laroque, BPA5, 98848, Nouméa cedex, New Caledonia
[3]Marine Biology Section, Department of Biology, University of Copenhagen, 3000 Helsingør, Denmark
[4]Sorbonne Universités, UPMC Université Paris 06, CNRS, Laboratoire d'Océanographie de Villefranche (LOV),
06230 Villefranche-sur-Mer, France
[5]Center for Prototype Climate Modeling, New York University Abu Dhabi, P.O. Box 129188, Abu Dhabi, United Arab
Emirates
[6]Department of Ecology, Environment, and Plant Sciences, Stockholm University, Stockholm Sweden 10690
*Correspondence to*: Sophie Bonnet (sophie.bonnet@univ-amu.fr)

31

32

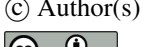



**Abstract**
Here we report quantification of $N_2$ fixation rates over a ~4000 km transect in the western and central tropical South
Pacific. Water samples were collected along a west to east transect from 160°E to 160°W, covering contrasting trophic
regimes, from oligotrophy in the Melanesian archipelagoes (MA) waters to ultra-oligotrophy in the South Pacific Gyre
(GY) waters. $N_2$ fixation was detected at all 17 sampled stations with an average rate of $631 \pm 286$ μmol N m$^{-2}$ d$^{-1}$
(range 196-1153 μmol N m$^{-2}$ d$^{-1}$) in MA waters and of $85 \pm 79$ μmol N m$^{-2}$ d$^{-1}$ (range 18-172 μmol N m$^{-2}$ d$^{-1}$) in GY
waters. Exceptionally high rates of $N_2$ fixation in MA waters were favored by availability of both iron and phosphate
and the observed warm sea surface temperatures (>28°C). *Trichodesmium* and UCYN-B cyanobacteria dominated the
diazotroph community (>80 %) and gene expression of nitrogenase genes (cDNA >$10^5$ *nifH* copies L$^{-1}$) in MA waters,
and single-cell isotopic analyses performed by nanoscale secondary ion mass spectrometry at selected stations reveal
that *Trichodesmium* was always the major contributor to $N_2$ fixation in MA waters, accounting for 47.1 to 83.8 % of
bulk $N_2$ fixation.










## 1 Introduction

In the ocean, nitrogen (N) availability in surface waters controls primary production and export of organic matter (Dugdale and Goering, 1967; Eppley and Peterson, 1979; Moore et al., 2013), a process commonly referred to as 'the biological carbon pump'. The major external source of new N to the ocean is biological $N_2$ fixation (100-150 Tg N $yr^{-1}$, (Gruber, 2008)), the conversion of $N_2$ gas dissolved in seawater into ammonia ($NH_4^+$). This process is performed by diazotrophic organisms possessing the enzyme nitrogenase encoded by the *nifH* genes. This source is continuously counteracted by N losses, mainly driven by denitrification and anammox, which convert fixed N (nitrate, $NO_3^-$) into $N_2$. Despite the critical importance of the N inventory in regulating primary production and export, the spatial distribution of N gains and losses in the ocean is still poorly resolved.

A global scale modeling study predicted that the highest rates of $N_2$ fixation are located in the South Pacific Ocean (Deutsch et al., 2007; Gruber, 2016). These authors also concluded that processes leading to N gains and losses are spatially coupled to oxygen deficient zones such as in the eastern tropical Pacific (ETSP), which harbors $NO_3^-$-poor but phosphate-rich surface waters, i.e. potentially ideal niches for $N_2$ fixation (Monteiro et al., 2011). However, recent field studies based on several cruises and independent approaches, including biological $^{15}N_2$ incubations-based measurements and geochemical $\delta^{15}N$ budgets, revealed low $N_2$ fixation rates (average range ~0-60 µmol N $m^{-2}$ $d^{-1}$) in the surface ETSP waters (Dekaezemacker et al., 2013; Fernandez et al., 2011; Fernandez et al., 2015; Knapp et al., 2016; Loescher et al., 2014), presumably due to iron (Fe) limitation (Bonnet et al., 2017; Dekaezemacker et al., 2013), as Fe is a major component of the nitrogenase enzyme (Raven, 1988). On the opposite, the western tropical South Pacific (WTSP) has recently been identified as a 'hot spot' of $N_2$ fixation (Bonnet et al., 2017) and together, these studies plead for a basin-wide spatial decoupling between $N_2$ fixation and denitrification in the South Pacific Ocean.

The WTSP is a vast oceanic region extending from Australia in the west to the western boundary of the South Pacific Gyre in the east (hereafter referred to as GY waters) (Figure 1). It has been chronically undersampled (Luo et al., 2012) compared to the tropical North Atlantic (Benavides and Voss, 2015) and North Pacific (e.g. (Böttjer et al., 2017)) oceans, but recent oceanographic surveys performed in the western part of the WTSP, in the Solomon, Bismarck (Berthelot et al., 2017; Bonnet et al., 2009; Bonnet et al., 2015) and Arafura (Messer et al., 2016; Montoya et al., 2004) Seas report extremely high $N_2$ fixation rates (>600 µmol N $m^{-2}$ $d^{-1}$, i.e. an order of magnitude higher than in the ETSP) throughout the year, that have been attributed to sea surface temperature >25°C and continuous nutrient inputs of terrigenous and volcanic origin (Labatut et al., 2014; Radic et al., 2011). The central and eastern parts of the WTSP, a vast oceanic region bordering Melanesian archipelagoes (New Caledonia, Vanuatu, Fiji) up to the Tonga trench (hereafter referred to as MA waters) have been far less investigated. One study (Shiozaki et al., 2014) reported high surface $N_2$-fixation rates close to Melanesian islands in relation with nutrient supplied by land runoff. However, the lack of direct $N_2$ fixation measurements over the full photic layer prevents the implementation of N budget estimates in this region. In addition, the reasons for such an ecological success of diazotrophs in the WTSP are still under debate (Bonnet et al., 2017) as the horizontal and vertical distribution of environmental parameters potentially controlling $N_2$ fixation, in particular Fe concentrations, are still scarce in this region.

Recurrent blooms of the filamentous cyanobacterium *Trichodesmium,* one of the most abundant diazotrophs in our oceans (Luo et al., 2012), have been reported in the WTSP since James Cook and Charles Darwin's expeditions and later confirmed by satellite observations (Dupouy et al., 2011; Dupouy et al., 2000) and microscopic enumerations



(Shiozaki et al., 2014; Tenorio et al., Accepted). However, molecular studies based on the *nifH* diversity also revealed
the presence of unicellular diazotrophic cyanobacteria (UCYN) in the WTSP (Moisander et al., 2010). Three groups
of UCYN (A, B and C) can be distinguished based on *nifH* gene sequences. In the warm (>25°C) waters of the Solomon
Sea, UCYN from group B (UCYN-B) co-occur with *Trichodesmium* at the surface, and together dominate the
diazotroph community (Bonnet et al., 2015), while UCYN-C are also occasionally abundant (Berthelot et al., 2017).
Further south in the Coral and Tasman Seas, UCYN-A dominate the diazotroph community (Bonnet et al., 2015;
Moisander et al., 2010). Both studies reported a transition zone from UCYN-B-dominated communities in warm
(>25°C) surface waters to UCYN-A-dominated communities in colder (<25°C) waters of the western part of the
WTSP. Further east in the MA waters, *Trichodesmium* and UCYN-B co-occur and account for the majority of total
*nifH* genes detected (Stenegren et al., 2017). Although molecular methods greatly enhanced our understanding of the
biogeographical distribution of diazotrophs in the WTSP, DNA-based *nifH* counts cannot be equaled to metabolic
activity. Thus, the contribution of each dominant group to bulk $N_2$ fixation is still lacking in this globally important
'hot spot' of $N_2$ fixation. Previous studies showed that different diazotrophs have different fates in the ocean: some are
directly exported, others release and transfer part of the recently fixed N to the planktonic food web and fuel indirect
export of organic matter (Berthelot et al., 2016; Bonnet et al., 2016a; Karl et al., 2012). Consequently assessing the
relative contribution of each dominating group of diazotrophs to overall $N_2$ fixation is critical to assess the
biogeochemical impact of $N_2$ fixation in the WTSP.

18         In the present study, we report previously undocumented bulk and group-specific $N_2$ fixation quantification

over a ~4000 km transect in the western and central tropical South Pacific. The goals of the study were i) to quantify
to horizontal and vertical distribution of $N_2$ fixation rates in the photic layer in relation with hydrological and biological
parameters, ii) to quantify the relative contribution of the dominant diazotrophs (*Trichodesmium* and UCYN-B) to $N_2$
fixation at selected stations based on cell-specific measurements, iii) to assess the potential ecological impact of $N_2$
fixation in this region.
**2       Methods**
Samples were collected during the 45-day OUTPACE (Oliotrophic to UlTra oligotrophic PACific Experiment) cruise
(DOI: http://dx.doi.org/10.17600/15000900) onboard the R/V *L'Atalante* in February-March 2015 (austral summer).
The west to east zonal transect along ~19°S started in Noumea (New Caledonia) and ended in Papeete (French
Polynesia) (Figure 1), covering contrasted trophic regimes (oligotrophy to ultra-oligotrophy) (see Moutin et al., 2017
for details), crossing MA waters around New Caledonia, Vanuatu, Fiji up to Tonga, and GY waters located at the
western boundary of the South Pacific Gyre. Data were collected at 17 stations including 14 short duration (SD; 8 h)
stations (SD1 to SD15, note that SD13 was not sampled) and three long duration (LD; 7 days) stations (LDA, LDB
and LDC). Vertical (0-200 m) profiles of temperature, salinity, and fluorescence were obtained at all 17 stations using
a Seabird 911 plus CTD equipped with a Wetlabs ECO-AFL/FL fluorometer. Seawater samples were collected by 12-
L Niskin bottles mounted on the CTD rosette.





## 2.1  Macro-nutrient and dissolved Fe concentrations analyses

Samples for quantifying nitrate ($NO_3^-$) and dissolved inorganic phosphorus (DIP) concentrations were collected at 12 depths between 0 and 200 m in acid-washed polyethylene bottles, fixed with $HgCl_2$ (final concentration 20 mg $L^{-1}$) and preserved at 4°C until analysis. Concentrations were determined using standard colorimetric techniques (Aminot and Kerouel, 2007) on a Bran Luebbe AA3 autoanalyzer. Detection limits for the procedures were 0.05 µmol $L^{-1}$ for $NO_3^-$ and DIP.

The sampling and analytical methods used to analyze the parameters reported in the correlation table (Table 2) are described in details in related papers in this issue (Bock et al., Submitted; Fumenia et al., Submitted; Stenegren et al., 2017; Van Wambeke et al., Submitted). Samples for dissolved Fe (DFe) concentrations determination were collected and analyzed as described in Guieu et al. (Under review).

## 2.2  Bulk N$_2$ fixation rate measurements

$N_2$ fixation rates were measured in triplicate at all 17 stations using the $^{15}N_2$ isotopic tracer technique (adapted from Montoya et al. (1996)). The $^{15}N_2$ bubble technique was intentionally chosen to avoid any potential overestimation due trace metal and dissolved organic matter (DOM) contaminations often associated with the preparation of the $^{15}N_2$-enriched seawater (Klawonn et al., 2015; Wilson et al., 2012) in our incubation bottles as Fe and DOM have been found to control $N_2$ fixation or *nifH* gene expression in this region (Benavides et al., 2017; Moisander et al., 2011). However, the $^{15}N/^{14}N$ ratio of the $N_2$ pool available for $N_2$ fixation (the term $A_{N2}$ used in the Montoya et al. (1996) equation) was measured in all incubation bottles to ensure accurate rate calculations (see below).

Seawater samples were collected in 10 % HCl-washed, sample-rinsed (3 times) light-transparent polycarbonate (2.3 L) bottles from 6 depths (75 %, 50 %, 20 %, 10 %, 1 %, and 0.1% surface irradiance levels) at short-duration stations SD1 to SD15 and 9 depths (75 %, 50 %, 35 %, 20 %, 10 %, 3 %, 1 %, 0.3 % and 0.1 % surface irradiance levels) at LDA, LDB and LDC, corresponding to the sub-surface (5 m) down to 80 to 180 m depending on the station. Bottles were sealed with caps fitted with silicon septa and amended with 2 mL of 98.9 atom% $^{15}N_2$ (Cambridge isotopes). The purity of the $^{15}N_2$ Cambridge isotopes stocks was previously checked by Dabundo et al. (2014) and more recently by (Benavides et al., 2015) and (Bonnet et al., 2016a). They were found to be lower than 2 x 10$^{-8}$ mol:mol of $^{15}N_2$, leading to a potential $N_2$ fixation rates overestimation of <1 %. Each bottle was agitated for 10 minutes to break the $^{15}N_2$ bubble and facilitate its dissolution and incubated for 24 h. At SD stations, bottles were incubated in on-deck incubators connected to circulating seawater at the specified irradiances using blue screening as the duration of the station (8 h) was too short to deploy in situ mooring lines. At LD stations (7 days), one profile was incubated following the same methodology in on-deck incubators and another replicate profile was incubated in situ for comparison on a drifting mooring line located at the same depth from which the samples were collected. Incubations were stopped by filtering the entire sample onto pre-combusted (450°C, 4 h) 25-mm diameter glass fiber filters (GF/F, Whatman, 0.7 µm nominal pore size), that were dried at 60°C for 24 h before being analyzed for $^{15}N/^{14}N$ ratios and particulate N (PN) concentrations determination using an elemental analyzer coupled to a mass spectrometer (EA-IRMS, Integra CN, SerCon Ltd) as described in (Bonnet et al., 2011).

To ensure accurate rate calculations, the $^{15}N/^{14}N$ ratio of the $N_2$ pool in the incubation bottles was measured on each profile from triplicate surface incubation bottles from SD1 to SD14 and at all depths at SD15 and LD stations.





Briefly, 12 mL were subsampled after incubation into Exetainers® fixed with $HgCl_2$ (final concentration 20 mg $L^{-1}$)
that were preserved upside down in the dark at 4°C until analyzed using a membrane inlet mass spectrometer (MIMS)
according to (Kana et al., 1994). Lastly, we collected time zero samples at each station to determine the natural N
isotopic signature of ambient particulate nitrogen (PN).

5        Discrete $N_2$ fixation rate measurements were depth integrated over the photic layer using trapezoidal

integration procedures assuming that surface $N_2$ fixation rates were identical to those in subsurface (5 m) and
considering that rates below the deepest sampled depth were zero (JGOFS, 1988).
**2.3    Statistical analyses**
Spearman's rank correlation was used to examine the potential relationships between $N_2$ fixation rates, hydrological,
biogeochemical, and biological parameters across the longitudinal transect (n=102, α=0.05). A non-parametric Mann-
Whitney test (α=0.05) was used to compare the MIMS data obtained following on-deck versus in situ incubations, and
to compare nutrient and Chl *a* distributions between the western part and the eastern part of the transect.
**2.4    Group-specific $N_2$ fixation rate measurements at selected stations**
**2.4.1    Experimental procedures**
At three selected stations along the transect (SD2, SD6, LDB) where *Trichodesmium* and UCYN-B accounted for >90
% of the total diazotrophic community (see below and Stenegren et al. (2017)), eight additional polycarbonate (2.3 L)
bottles were collected from the surface (50 % light irradiance) to determine *Trichodesmium* and UCYN-B specific $N_2$
fixation rates by nanoSIMS and quantify their contribution to bulk $N_2$ fixation. Two of them were amended with $^{15}N_2$
as described above for further nanoSIMS analyses on individual cells (the 6 remaining bottles were used for DNA and
RNA analyses, see below) and were incubated for 24 h with the incubation bottles dedicated to bulk $N_2$ fixation
measurements in on-deck incubators as described above. To recover large-size diazotrophs (*Trichodesmium*) after
incubation, 1.5 L were filtered on 10 μm pore size 25 mm diameter polycarbonate filters. The cells were fixed with
paraformaldehyde (PFA) (2 % final concentration) for 1 h at ambient temperature (~25 °C) and the filters were then
stored at -20°C until nanoSIMS analyses. To recover small size diazotrophs (UCYN-B), samples were collected for
further cell sorting by flow cytometry prior to nanoSIMS. 1 L of the remaining $^{15}N_2$ labelled bottle were filtered onto
0.2 μm pore size 47 mm polycarbonate filters. Filters were quickly placed in a 5 mL cryotube® filled with 0.2 μm
filtered seawater with PFA (2 % final concentration) for 1 h at room temperature in the dark. The cryovials were
vortexed for 10 s to detach the cells from the filter (Thompson et al., 2012) and stored at -80°C until cell sorting. Cell
sorting of UCYN-B was performed on a Becton Dickinson Influx™ Mariner (BD Biosciences, Franklin Lakes, NJ)
high speed cell sorter of the Regional Flow Cytometry Platform for Microbiology (PRECYM), hosted by the
Mediterranean Institute of Oceanography, as described in Bonnet et al. (2016a) and (Berthelot et al., 2016). After
sorting, the cells were dropped onto a 0.2 μm pore size polycarbonate 13 mm diameter polycarbonate filter connected
to low pressure vacuum pump, then stored at -80°C until nanoSIMS analyses. Special care was taken to drop the cells
on a surface as small as possible (~5 mm in diameter) to ensure the highest cell density possible to facilitate subsequent
nanoSIMS analyses.



### 2.4.2    Abundance of diazotrophs and *nifH* gene expression

The abundance of *Trichodesmium* filaments and the average number of cells/filament was determined microscopically: 1 to 2.2 L were filtered on 2 µm polycarbonate filters. The cells were fixed with PFA (2 % final concentration) for 1 h at 4°C and stored at -20°C until counting using an epifluorescence microscope (Zeiss Axioplan, Jana, Germany) fitted with a green (510–560 nm) excitation filter.

Four other diazotrophic phylotypes were quantified using quantitative PCR (qPCR) as they were too scarce or not enumerable by microscopy: UCYN-A1, UCYN-B and two heterocystous symbionts of diatom-diazotroph associations (DDAs): *Richelia intracellularis* associated with *Rhizosolenia* spp. (het-1) and *Richelia intracellularis* associated with *Hemiaulus* spp. (het-2). Triplicate 2.3 L-bottles were filtered onto 25 mm diameter 0.2 µm Supor filters with a 0.2 µm pore size at each station using a peristaltic pump. The DNA extraction and TaqMAN qPCR assays are fully described in (Stenegren et al., 2017). To evaluate the *nifH* gene expression, additional triplicate 2.3 L bottles were filtered as described above. The filters were placed into pre-sterilized bead-beater tubes (Biospec Products Inc., Bartlesville, OK USA) containing 250 µL RLT buffer (Qiagen RNeasy) amended with 1 % ß-mercaptoethanol and 30 µL of 0.1 mm glass beads (Biospec Products Inc.). The time of filtering for RNA varied between stations (17-21:00). Filters were flash frozen in liquid nitrogen and stored at -80 C until RNA extraction. The RNA extraction and Reverse Transcription were performed as previously described using a Super-Script III first-strand cDNA synthesis kit (Invitrogen Corp., Carlsbad, CA, USA) including the appropriate negative controls (water, and No RT) (Foster et al., 2010).

### 2.4.3    nanoSIMS analyses, data processing and group-specific rate calculations

NanoSIMS analyses were performed using a nanoSIMS N50 at the French National Ion MicroProbe Facility according to (Bonnet et al., 2016a; Bonnet et al., 2016b) and (Berthelot et al., 2016). Briefly, a ~ 1.3 pA Cesium (16KeV) primary beam focused onto ~100 nm spot diameter was scanned across a 256 x 256 or 512 x 512 pixel raster (depending on the image size) with a counting time of 1 ms per pixel. Samples were pre-sputtered prior to analyses with a current of ~10 pA for at least 2 min to achieve sputtering equilibrium and ensure a consistent implantation and analysis of the cell interior by removing cell surface. Negative secondary ions ($^{12}C^{14}N^-$, $^{12}C^{15}N^-$) were collected by electron multiplier detectors, and secondary electrons were also imaged simultaneously. A total of 10-50 serial quantitative secondary ion images were generated, that were combined to create the final image. Mass resolving power was ~8000 in order to resolve isobaric interferences. 20 to 100 planes were generated for each cells analyzed. NanoSIMS runs are time intensive and not designed for routine analysis, but a minimum of 250 cells of UCYN-B per station and 30 *Trichodesmium* filament portions were analyzed to take into account the variability of activity among the population.

Data were processed using the LIMAGE software. Briefly, all scans were corrected for any drift of the beam and sample stage during acquisition. Isotope ratio images were created by adding the secondary ion counts for each recorded secondary ion for each pixel over all recorded planes and dividing the total counts by the total counts of a selected reference mass. Individual *Trichodesmium* filaments and UCYN-B cells were easily identified on nanoSIMS images that were used to define regions of interest (ROIs). For each ROI, the $^{15}N/^{14}N$ ratio was calculated.

*Trichodesmium* and UCYN-B cellular biovolume was calculated from cell-diameter measurements performed on 50 ~cells or trichomes per station on an epifluorescence microscope (Zeiss Axioplan, Jana, Germany) fitted with a



green (510–560 nm) excitation filter. UCYN-B had a spherical shape and *Trichodesmium* cells were assumed to have
a cylindrical shape. The carbon content per cell was determined from the biovolume according to Verity et al. (1992)
and the N content was calculated based on a C:N ratios of 6 for *Trichodesmium* (Carpenter et al., 2004) and 5 for
UCYN-B (Dekaezemacker and Bonnet, 2011; Knapp et al., 2012). $^{15}$N assimilation rates were expressed 'par cell' and
calculated as follows (Foster et al., 2011; Foster et al., 2013): assimilation (mol N cell$^{-1}$ d$^{-1}$) = ($^{15}$Nex x N$_{con}$)/N$_{sr}$, where
$^{15}$Nex is the excess $^{15}$N enrichment of the individual cells measured by nanoSIMS after 24 h of incubation relative to
the time zero value, N$_{con}$ is the N content of each cell determined as described above, and N$_{sr}$ is the excess $^{15}$N
enrichment of the source pool (N$_2$) in the experimental bottles determined by MIMS (see above). Standard deviations
were calculated using the variability of N isotopic signature measured by nanoSIMS on replicate cells. The relative
contribution of *Trichodesmium* and UCYN-B to bulk N$_2$ fixation was calculated by multiplying cell-specific N
assimilation by the cell abundance of each group, relative to bulk N$_2$ fixation determined at the same time.
**3    Results**
**3.1    Environmental conditions**
Seawater temperature ranged from 21.4 to 30.0 °C in the sampled photic layer (0 to ~80-180 m) over the cruise transect
(Figure 2a). Maximum temperatures were measured at the surface (0-50 m, 28.7 °C on average) and remained constant
along the longitudinal transect, with one exception when slightly higher sea surface temperatures (SST) were observed
at LDB (29.9°C) compared to the transect average SST (28.7°C).
For nutrients, DFe and Chl *a* concentrations, the transect was divided into two main characteristic sub-regions:
1) the MA region from station SD1 (160°E) to LDB (165°W), and 2) the GY sub-region from station LDB (165°W)
to SD15 (160°W) located in the South Pacific Gyre. Chl *a* concentrations were significantly (p<0.05) higher in MA
waters (0.17 µg L$^{-1}$ on average over 0-50 m) than in GY waters (0.06 µg L$^{-1}$ on average over 0-50 m) (Figure 1, Figure
2b). The DCM was located around 80-100 m in MA waters and deepened at ~150 m in GY waters. Surface NO$_3^-$
concentrations (Figure 2c) were close or below the detection limit (0.05 µmol L$^{-1}$) all over the surface (0-50 m) waters
throughout the transect, but the depth of the nitracline gradually deepened from ~80-100 m in MA waters down to
~150 m in GY waters. DIP concentrations were slightly higher or close to detection limits (0.05 µmol L$^{-1}$) in MA
surface (0-50 m) waters and increased significantly (p<0.05) in GY waters to reach 0.13-0.17 µmol L$^{-1}$ (Figure 2d).
**3.2    N isotopic signature of the N$_2$ pool after incubation (MIMS results)**
The $^{15}$N enrichment of the N$_2$ pool after 24 h of incubation with the $^{15}$N$_2$ tracer was on average 6.145 ± 0.798 atom%
(n=54) in bottles incubated in on-deck incubators and significantly higher (p<0.05) in bottles incubated on the in situ
mooring line (7.548 ± 0.557 atom% (n=44), Figure 3a). However, the depth of incubation on the in situ mooring line
(between 5 and 180 m) did not have any significant effect (p>0.05) on the isotopic signature of the N$_2$ pool at LDB
and LDC, which remained constant over the water column (Figure 3b).

**3.3    Natural isotopic signature of suspended particles and N$_2$ fixation rates**
The $^{15}$N/$^{14}$N ratio of suspended particles measured over the photic layer was on average -0.41‰ in MA waters and
8.06‰ in GY waters (Table 1). Those numbers were used as time zero samples to calculate N$_2$ fixation rates.





N$_2$ fixation was detected at all 17 sampled stations and the transect could also be divided into the two main
characteristic sub-areas: 1) the MA waters exhibiting average N$_2$ fixation rates of 8.9 ± 10 nmol N L$^{-1}$ d$^{-1}$ (range: DL-
48 nmol N L$^{-1}$ d$^{-1}$) over the photic layer, and 2) the GY waters exhibiting average N$_2$ fixation rates of 0.5 ± 0.6 nmol
N L$^{-1}$ d$^{-1}$ (range: 0-4.0 nmol N L$^{-1}$ d$^{-1}$) (Figure 2e). In MA waters, N$_2$ fixation was mostly restricted to the surface (0-
25 m), where rates commonly peaked at 30 to 48 nmol N L$^{-1}$ d$^{-1}$. In GY waters, maximum rates reached 1-2 nmol N
L$^{-1}$ d$^{-1}$ and were located deeper in the water column (~50 m). When integrated over the photic layer, N$_2$ fixation
represented an average net N addition of 631 ± 286 µmol N m$^{-2}$ d$^{-1}$ (range 196-1153 µmol N m$^{-2}$ d$^{-1}$) in MA waters
and of 85 ± 79 µmol N m$^{-2}$ d$^{-1}$ (range 18-172 µmol N m$^{-2}$ d$^{-1}$) in GY waters.
**3.4      Correlations between N$_2$ fixation and hydrological, biogeochemical and biological parameters**
N$_2$ fixation rates were significantly positively correlated with seawater temperature and photosynthetically active
radiation (PAR) ($p<0.05$) and significantly negatively correlated with depth and salinity ($p<0.05$) and not significantly
correlated with dissolved oxygen concentrations ($p>0.05$) (Table 2). Regarding the main biogeochemical stocks and
fluxes measured during the cruise, N$_2$ fixation rates were significantly positively correlated with dissolved organic N
(DON), phosphorus (DOP) and carbon (DOC), particulate organic N (PON), particulate organic carbon (POC),
biogenic silica (BSi) and Chl $a$ concentrations ($p<0.05$) and significantly negatively correlated with NO$_3^-$, NH$_4^+$, DIP
and silicate concentrations ($p<0.05$).
In terms of diazotrophic groups based on the quantification of *nifH* genes by qPCR (Stenegren et al., 2017),
N$_2$ fixation rates were significantly positively correlated with *Trichodesmium* spp., UCYN-B and the three DDAs (het-
1, het-2 and het-3) abundances ($p<0.05$) and not significantly correlated with UCYN-A1 and UCYN-A2 abundances
($p>0.05$) (Table 2). Regarding non-diazotrophic plankton determined by flow cytometry (Bock et al., Submitted), N$_2$
fixation rates were significantly positively correlated with *Prochlorococcus* spp., *Synechococcus* spp., heterotrophic
bacteria and protists abundances ($p<0.05$) and significantly negatively correlated with picoeukaryotes ($p<0.05$).
**3.5      Contribution of *Trichodesmium* and UCYN-B to N$_2$ fixation and nitrogenase gene expression**
At the three stations where cell specific N$_2$ fixation rates were estimated by nanoSIMS (SD2, SD6 and LDB), the most
abundant diazotroph phylotype was *Trichodesmium* with 1.3 x 10$^5$, 3.3 x 10$^5$ and 1.2 x 10$^5$ cells L$^{-1}$ respectively,
followed by UCYN-B, which abundances were 2.0 x 10$^4$, 1.5 x10$^5$ and 3.8 x 10$^2$ *nifH* copies L$^{-1}$ respectively. Het-1
and het-2 combined were one to two orders of magnitude lower, ranging from 1.0 to 9.9 x 10$^3$ *nifH* copies L$^{-1}$ and
UCYN-A1 were below detection at the three stations. In summary, *Trichodesmium* and UCYN-B accounted for 98.2
%, 99.8 % and 92.1 % of the total diazotroph community (based on the phylotypes targeted here) at SD2, SD6 and
LDB, respectively (Table 3).
The $^{15}$N/$^{14}$N ratio of individual cells/trichomes of UCYN-B and *Trichodesmium* were measured via nanoSIMS
analyses and used to estimate single cell N$_2$ fixation rates. A summary of the enrichment values and cell-specific N$_2$
fixation is provided in Table 3. Individual trichomes exhibited significant $^{15}$N enrichments (0.610 ± 0.269, 0.637 ±
0.355 and 0.981 ± 0.466 atom% at stations SD2, SD6 and LDB, respectively) compared with time zero samples (0.369
± 0.002 atom%). UCYN-B were also significantly $^{15}$N-enriched with 1.163 ± 0.531 atom% and 0.517 ± 0.237 atom%
at SD2 and SD6, respectively (no UCYN-B could be sorted and analyzed by nanoSIMS at LDB as they accounted for





only 0.3 % of the diazotroph community). Cell-specific $N_2$ fixation of *Trichodesmium* were $38.9 \pm 8.1$, $29.3 \pm 5.4$ and
$123.8 \pm 24.8$ fmol N cell $d^{-1}$ at SD 2, SD6 and LDB.  Cell-specific $N_2$ fixation of UCYN-B were $30.0 \pm 6.4$ and $6.1 \pm$
$1.2$ fmol N cell $d^{-1}$ at SD1 and SD6. The contribution of *Trichodesmium* to bulk $N_2$ fixation was 83.8 %, 47.1 % and
52.9 % at stations SD2, SD6 and LDB, respectively. The contribution of UCYN-B was 10.1 %, 6.1 % at SD2 and SD6,
respectively (Table 3).

6         The *in situ nifH* expression for all diazotroph groups targeted by qPCR was estimated using a TaqMAN

quantitative reverse transcription PCR (RT-QPCR) (Table 4).  The sampling and filtering time (17-21:00 h) was not
optimal for quantifying the *nifH* gene expression for all diazotrophs, however it is useful evaluation for which
diazotrophs were active during the experiment. Both *Trichodesmium* and UCYN-B dominated the biomass (Stenegren
et al., 2017), as did their *nifH* gene expression at all three stations, especially SD2 and SD6. Of the two het-groups,
het-1 had a higher *nifH* gene expression, which was consistent with its higher *nifH* abundance by standard qPCR
(Stenegren et al., 2017).  UCYN-A1 was consistently below detection for the *nifH* gene expression and was also the
least detected diazotroph by *nifH* qPCR.
**4      Discussion**
**4.1      Methodological considerations: the importance of measuring the $^{15}N/^{14}N$ ratio of the $N_2$ pool**
Our understanding of the marine N cycle relies on accurate measurements of N fluxes to and from the ocean. Two
methods are routinely used by the scientific community to perform direct $N_2$ fixation measurements in marine systems:
1) the method developed by (Montoya et al., 1996), which consists of the addition of the $^{15}N_2$ tracer as a bubble in the
incubation bottles (hereafter referred to as the 'bubble addition method') and the measurement of the $^{15}N/^{14}N$ ratio of
PN before (time zero) and after incubation, 2) the method consisting of adding the $^{15}N_2$ as dissolved in a subset of
seawater previously $N_2$ degassed (Mohr et al., 2010) (hereafter referred to as the '$^{15}N_2$-enriched seawater method').
The second method was developed because the first had been observed to potentially underestimate $N_2$ fixation rates
(Großkopf et al., 2012; Mohr et al., 2010; Wilson et al., 2012) due to the incomplete (and gradually increasing during
the incubation period) equilibration of the $^{15}N_2$ in the incubation bottles when injected as a bubble. This results in a
lower $^{15}N/^{14}N$ ratio of the $N_2$ pool available for $N_2$ fixation (the term $A_{N2}$ used in the Montoya et al. (1996) equation)
as compared to the theoretical value calculated based on gas constants, and therefore potentially leads to
underestimated rates in some studies (see references above), whereas other studies do not see any significant
differences between both methods (Bonnet et al., 2016c; Shiozaki et al., 2015). Here we decided to use the 'bubble
addition method' to minimize potential trace metal and organic matter contaminations, which may have resulted in
overestimating rates (Klawonn et al., 2015). However, we paid careful attention to accurately measure the term $A_{N2}$ to
avoid any potential underestimation and reveal that the way bottles are incubated (on-deck *versus* in situ) has a great
influence of the $A_{N2}$ value, and thus on $N_2$ fixation rates.

34         Our MIMS results reveal a significantly ($p<0.05$) lower $^{15}N$ enrichment of the $N_2$ pool ($6.145 \pm 0.798$ atom%)

when bottles were incubated in on-deck incubators compared to when bottles were incubated on the in situ mooring
line ($7.548 \pm 0.557$ atom%). This suggests that the $^{15}N_2$ dissolution is much more efficient when bottles are incubated
in situ, probably due to the higher pressure in seawater at the depth of incubation (1.5 to 19 bars between 5 and 180
m) compared to the pressure in the on-deck incubators (1 bar). The seawater temperature checked regularly in the on-





deck incubators was equivalent to ambient SST and likely did not explain the differences observed. This result
highlights the need to perform systematic MIMS measurements to use the most accurate $A_{N2}$ value for rate calculations,
independently of the $^{15}N_2$ approach used (gas or dissolved). In our study, the theoretical $A_{N2}$ value based on gas
constants calculations (Weiss, 1970) was ~8.2 atom%, so the deviation from this value is more important when bottles
are incubated in on-deck incubators as compared to when they are incubated in situ. This suggests that the use of the
bubble addition method without MIMS measurement potentially leads to higher underestimations when bottles are
incubated in on-deck incubators, which is the case in the great majority of marine $N_2$ fixation studies published so far
(Luo et al., 2012). We are aware the dissolution kinetics of $^{15}N_2$ in the incubation bottles may have been progressive
along the 24 h of incubation (Mohr et al., 2010), therefore, the $N_2$ fixation rates provided here represent conservative
values.
Despite the $A_{N2}$ value differed according to the incubation mode, it did not change with the depth of incubation
on the mooring line, indicating that a slightly higher pressure than atmospheric pressure (1.5 bar at 5 m depth) is
enough to promote the $^{15}N_2$ dissolution. In our study, the vertical profiles performed at LD stations and incubated either
on-deck in triplicate or in situ in triplicates reveal identical ($p>0.05$) $N_2$ fixation rates regardless of the incubation
method used (Caffin et al., 2017). This indicates that in situ incubations and under in situ-simulated conditions (on-
deck incubators) is a valid methodology for $^{15}N_2$ rate measurements on cruises during which in situ mooring lines
cannot be deployed, as long as systematic measurements of the isotopic ratio of the $N_2$ pool is performed in incubation
bottles.
**4.2      What causes such a hot spot of $N_2$ fixation in the WTSP?**
$N_2$ fixation rates measured in MA waters (average $631 \pm 286$ µmol N m$^{-2}$ d$^{-1}$) are three to four times higher than model
predictions for this area (150-200 µmol N m$^{-2}$ d$^{-1}$, Gruber (2016)). They are in the upper range of the upper category
(100-1000 µmol N m$^{-2}$ d$^{-1}$) of rates defined by (Luo et al., 2012) in the global $N_2$ fixation MAREDAT database and
thus reveal the WTSP as a 'hot spot' of $N_2$ fixation in the global ocean. Recent studies performed in the western part
of the WTSP, ie. in the Solomon, Bismarck (Berthelot et al., 2017; Bonnet et al., 2009; Bonnet et al., 2015) and Arafura
(Messer et al., 2015; Montoya et al., 2004) Seas also reveal extremely high rates (>600 µmol N m$^{-2}$ d$^{-1}$), indicating that
this 'hot spot' of $N_2$ fixation extends geographically west-east from Australia to Tonga and north-south from the
equator to 25-30°S, covering a vast ocean area of ~13 x $10^6$ km$^2$ , (i.e. ~20 % of the South Pacific Ocean area).
However, the reasons for such an ecological success of diazotrophs in this region are still poorly understood and raise
the question of 'which factors influence the distribution and activity of $N_2$ fixation in the ocean?' In a study conducted
in 2014 at global scale, Luo et al. (2014) investigated the statistical links between the spatial variation of $N_2$ fixation
and that of environmental parameters commonly accepted to control this process: surface $NO_3^-$ and DIP concentrations,
the tracer P*, atmospheric deposition, sea surface temperature (SST), mixed layer depth, solar radiation in the mixing
layer, wind speed and minimum oxygen concentration in the 0-500 m layer. They concluded that the best predictor to
explain the spatial distribution of $N_2$ fixation in the ocean is SST (or surface solar radiation). Below we highlight the
most plausible factors explaining this 'hot spot' of $N_2$ fixation.
**SST**. SST was unlikely the factor explaining the differences in $N_2$ fixation rates observed between MA and
GY waters, as SST was consistently high (28.7°C on average over the 0-50 m layer) and optimal for the growth and





nitrogenase activity of most diazotrophs (Breitbarth et al., 2007; Fu et al., 2014) all along the cruise transect. This
indicates that other factors such as nutrient availability may explain the distribution of $N_2$ fixation.
**DIP availability**. The ~4000 km transect was clearly divided into two main sub-regions: 1) the MA waters,
harboring typical oligotrophic conditions with surface (0-50 m) $NO_3^-$ and DIP concentrations close to detection limits
(0.05 μmol $L^{-1}$), moderate surface (0-50 m) Chl $a$ concentrations (~0.17 μg $L^{-1}$), a DCM located at ~80-100 m and
very high $N_2$ fixation rates (631 ± 286 μmol N $m^{-2}$ $d^{-1}$ on average), 2) the GY waters harboring ultra-oligotrophic
conditions with undetectable $NO_3^-$, high DIP concentrations (0.15 μmol $L^{-1}$), very low Chl $a$ concentrations (0.06 μg
$L^{-1}$, DCM ~150 m) and low $N_2$ fixation rates (85 ± 79 μmol N $m^{-2}$ $d^{-1}$).
In the $NO_3^-$-depleted MA waters, the DIP concentrations close to the detection limit are indicative of the
consumption of DIP by diazotrophs. This is consistent with the negative correlation found between $N_2$ fixation and
DIP turn-over time ($T_{DIP}$, the ratio between DIP concentrations and DIP uptake rates) during the OUTPACE cruise
(Table 2), indicative of higher DIP limitation when $N_2$ fixation increases and consume DIP. On the opposite, DIP
concentrations are high (>0.1 μmol $L^{-1}$) in GY surface waters, consistent with former studies considering the South
Pacific Gyre as a High Phosphate, Low Chlorophyll ecosystem (Moutin et al., 2008), in which DIP accumulates in the
absence of $NO_3^-$ and low $N_2$ fixation activity, which is suspected to be limited by temperature and/or Fe availability
(Bonnet et al., 2008; Moutin et al., 2008). Moutin et al., (2005) have shown that seasonal variations in DIP availability
control the growth and decline of *Trichodesmium* blooms in New Caledonian waters. During the OUTPACE cruise,
$T_{DIPs}$ were variable in MA waters but always much higher than those typically measured in severely DIP-limited
environments such as the Mediterranean and the Sargasso Seas (e.g. (Moutin et al., 2008)), suggesting that DIP
concentrations are generally favorable for the development of diazotrophs in the WTSP, and do not alone explain why
$N_2$ fixation is high in MA waters and low in GY waters. However, it is likely that the depletion of DIP stocks at the
end of the summer season forces the decline of diazotrophic blooms in the WTSP (Moutin et al., 2005), concomitantly
with the decline in SST.
**Fe availability**. Before OUTPACE, our knowledge on Fe sources and concentrations in the WTSP was
patchy, especially in MA waters. During OUTPACE, Guieu et al. (Under review) reported high DFe concentrations in
MA waters (1.7 nmol $L^{-1}$ on average over the photic layer), i.e significantly (p<0.05) higher than those reported in GY
waters (0.3 nM on average over the photic layer). The low DFe concentrations measured in the GY waters are in
accordance with previous reports for the same region (Blain et al., 2008; Fitzsimmons et al., 2014). However, the high
DFe concentrations measured in MA waters were previously undocumented, and reveal several maxima (> 50 nmol
$L^{-1}$) between stations SD7 to SD11, suggesting intense fertilization processes taking place in this region. Guieu et al.
(Under review) found that atmospheric deposition in this region was too low to explain the observed DFe
concentrations in the water column, and that the Fe inputs up to the euphotic layer is from a shallow (~500 m)
hydrothermal source. The seafloor of the WTSP hosts the Tonga-Kermadec subduction zone which stretches 2500 km
from New Zealand to the Tonga archipelago. It has among the highest density of submarine volcanoes associated with
hydrothermal vents recorded in the ocean (2.6 vents/100 km; Massoth et al. (2007)), which discharge large quantities
of material into the water column, including biogeochemically relevant elements such as Fe, manganese, etc. By use
of  modeling simulations, Guieu et al., (Under review) hypothesize that such shallow Fe sources could spread
throughout the WTSP through mesoscale activity and mainly predominant westward currents such as the South



Equatorial Current, SEC (Figure 1) and thus explain the high DFe concentrations in MA waters compared to the GY
ones. In our study, DFe concentrations were significantly positively correlated with $N_2$ fixation, likely contributing to
explaining the contrasted $N_2$ fixation rates observed across the OUTPACE transect. This is in accordance with recent
model simulations performed at the Pacific scale, which show that deep Fe sources controls the spatial distribution and
the abundance of *Trichodesmium* in the WTSP (Dutheil et al., Submitted).

6       Our hypothesis to explain the spatial distribution of $N_2$ fixation in this region is the following: when DIP-rich

waters flow westward from the ETSP through the SEC and cross the South Pacific Gyre, $N_2$-fixing organisms do not
develop despite optimal SST (>25°C), likely because GY waters are Fe-depleted (Bonnet et al., 2008; Moutin et al.,
2008). When these DIP-rich waters pass the Tonga trench, the high DFe concentrations associated with SST >25°C
altogether would provide ideal conditions for diazotrophs to bloom extensively and likely explain the 'hot spot' of $N_2$
fixation in the region. Further investigations are required to better quantify Fe input from shallow volcanoes and
associated hydrothermal activity along the Tonga volcanic arc for the upper mixed layer, study the fate of hydrothermal
plumes in the water column at the local and regional scales, and investigate the potential impact of such hydrothermal
inputs on diazotrophic communities at the scale of the whole WTSP.

15       Besides DFe, $N_2$ fixation rates were significantly negatively correlated with depth, which likely explains the

significantly positive correlations between $N_2$ fixation and PAR and $N_2$ fixation and seawater temperature, those two
parameters being also depth dependent (the thermocline was roughly located around 50 m). $N_2$ fixation rates were the
highest were $NO_3^-$ concentrations were the lowest, and both were significantly negatively correlated, consistent with
the high energetic cost of $N_2$ fixation compared to $NO_3^-$ assimilation (Falkowski, 1983).
**4.3    *Trichodesmium*: the major contributor to $N_2$ fixation in the WTSP**
In this 'hot spot' of $N_2$ fixation (MA waters), the dominant diazotroph phylotypes quantified using *nifH* quantitative
PCR assays were *Trichodesmium* spp. and UCYN-B (Stenegren et al., 2017), which commonly peaked at >$10^6$ *nifH*
copies $L^{-1}$ in surface (0-50 m) waters. DDAs (mainly het-1, but het-2 and het-3 were also detected) were the next most
abundant diazotrophs (Stenegren et al., 2017). This result is consistent with the fact that abundances of those four
phylotypes co-varied and were significantly positively correlated with $N_2$ fixation rates (Table 2). The two UCYN-A
lineages (UCYN-A1 and UCYN-A2) were less abundant (<1.0-1.5 % of total *nifH* copies, Stenegren et al., 2017) and
not significantly correlated with $N_2$ fixation rates (Table 2).

29       The relative contribution of different diazotroph phylotypes to bulk $N_2$ fixation has been largely investigated

through bulk and size fractionation measurements (usually comparing > and <10 μm size fraction $N_2$ fixation rates),
which may be misleading since some small-size diazotrophs are attached to large-size particles (Benavides et al., 2016;
Bonnet et al., 2009) and some diazotrophic-derived N released by diazotrophs is assimilated by small and large non-
diazotrophic plankton (e.g. Bonnet et al. (2016a)). Here we directly measure the *in situ* cell-specific $N_2$ fixation activity
of the two diazotroph groups dominating the community in MA waters: *Trichodesmium* and UCYN-B.

35       At all three studied stations, *Trichodesmium* was dominating, accounting for 68.0-91.8 % of the diazotroph

community, followed by UCYN-B, accounting for 0.3 to 31.7 %. Likewise *Trichodesmium* and UCYN-B had the
highest gene expression ($10^2$-$10^5$ cDNA *nifH* copies $L^{-1}$). It was not surprising that UCYN-B had high gene expression
given that the sampling time occurred later in the day (17-21:00), however both *Trichodesmium* and het-1 which



typically reduce $N_2$ and express *nifH* highest during the day (Church et al. 2005), had detectable and often equally high
expression as UCYN-B. Cell-specific $N_2$ fixation rates reported here are in the same order of magnitude as those
reported for field populations of *Trichodesmium* (Berthelot et al., 2016; Martinez-Perez et al., 2016) and UCYN-B
(Foster et al., 2013). *Trichodesmium* was always the major contributor to $N_2$ fixation, accounting for 47.1 to 83.8 % of
bulk $N_2$ fixation, while UCYN-B never exceeded 6.1-10.1 %, despite accounting for >30 % of the diazotroph
community at SD6. This may be linked with the lower $^{15}N$ enrichment at SD6 ($0.517 \pm 0.237$ atom%), which is due to
a high proportion of inactive cells (atom% close to natural abundance) compared to SD2, where the majority of cells
were active and highly $^{15}N$-enriched ($1.163 \pm 0.531$ atom%). Such heterogeneity in $N_2$ fixation rates among cells has
already been reported by Foster et al., (2013). Overall, these results show that the most abundant phylotype
(*Trichodesmium*) accounts for the majority of $N_2$ fixation, but not in the same proportion, highlighting that the
abundance of micro-organisms in seawater cannot be equated to activity, which has already been reported for other
functional groups such as bacteria (Boutrif et al., 2011). In the North Pacific Gyre (Station ALOHA), Foster et al.
(2013) report a higher contribution of UCYN-B to bulk $N_2$ fixation (24-63 %) during the summer season, indicating
that this group likely contributes more to the N budget at station ALOHA than in the WTSP, where *Trichodesmium*
seems to be the major player.
**4.4     Ecological relevance of $N_2$ fixation in the WTSP**
$N_2$ fixation was significantly positively correlated with Chl *a*, PON, POC and BSi concentrations, as well as with
primary production, suggesting a tight coupling between $N_2$ fixation, primary production and biomass accumulation
in the water column. Based on our measured C:N ratios at each depth, the computation of the N demand derived from
primary production measured during OUTPACE (Van Wambeke et al., Submitted) indicates that $N_2$ fixation fueled on
average $8.2 \pm 1.9$ % (range 5.9 to 11.5 %) of total primary production in the WTSP. This contribution is higher than
in other oligotrophic regions such as the Northwestern Pacific (Shiozaki et al., 2013), ETSP (Raimbault and Garcia,
2008), Northeast Atlantic (Benavides et al., 2013b) or the Mediterranean Sea (Bonnet et al., 2011; Ridame et al., 2013),
where it is generally <5 %. However, it is comparable to results found further North in the Solomon Sea ($N_2$ fixation
fueled 9.4 % of primary production, Berthelot et al. (2017)), which is part of the WTSP 'hot spot' of $N_2$ fixation
(Bonnet et al., 2017). Caffin et al. (2017) show that $N_2$ fixation represents the major source (>90 %) of new N to the
upper photic (productive) layer during the OUTPACE cruise, before atmospheric inputs and nitrate diffusion across
the thermocline, indicating that $N_2$ fixation supported nearly all new production in this region during austral summer
conditions.

31        The large amount of N provided by $N_2$ fixation likely stimulated the growth of non-diazotrophic plankton

as suggested by significant positive correlations between $N_2$ fixation rates and the abundance of *Prochlorococcus* spp.,
*Synechococcus* spp., heterotrophic bacteria and protists. $^{15}N_2$ based transfer experiments coupled with nanoSIMS
experiments designed to trace the passage of $^{15}N$ in the planktonic food web demonstrated that ~10 % of diazotroph-
derived N is rapidly (24-48 h) transferred to non-diazotrophic phytoplankton (mainly diatoms and bacteria) in coastal
waters of the WTSP (Bonnet et al., 2016a,b; Berthelot et al., 2016). The same experiments performed in offshore
waters during the present cruise confirm that ~10 % of recently-fixed $N_2$ are also transferred to picophytoplankton and
bacteria after 48 h (Caffin et al., Submitted). This is in accordance with Van Wambeke et al., Submitted) who report





that $N_2$ fixation fuels 30 to 70 % of the bacteria N demand in MA waters. This further demonstrates that $N_2$ fixation
acts as an efficient natural N fertilization in the WTSP, potentially fueling subsequent export of organic material below
the photic layer. Caffin et al. (2017) estimated that the $e$-ratio, which quantifies the efficiency of a system to export
particulate carbon relative to primary production ($e$-ratio = POC export/PP), was three times higher (p<0.05) in MA
waters compared to GY waters. Moreover, $e$-ratio values were as high as 9.7 in MA waters, i.e. higher than the $e$-ratios
in most studied oligotrophic regions (Karl et al., 2012; Raimbault and Garcia, 2008), where it rarely exceed 1 %,
indicating that production sustained by $N_2$ fixation is efficiently exported in the WTSP. Diazotrophs were recovered
in sediment traps during the cruise (Caffin et al., 2017), but their biomass only accounted for ~5 % (locally 30% at
LDA) of the N biomass in the traps, indicating that most of the export was indirect, i.e. after transfer of diazotroph-
derived N to the surrounding planktonic communities that were subsequently exported. A $\delta^{15}$N-budget performed
during the OUTPACE cruise reveals that $N_2$ fixation supports exceptionally high (>50 % and locally >80 %) of export
production in MA waters (Knapp et al., Submitted). Together these results suggest that $N_2$ fixation plays a critical role
in export in this globally important 'hot spot' of $N_2$ fixation.
**5      Conclusions**
The magnitude and geographic extent of $N_2$ fixation control the rate of primary productivity and vertical export of
carbon in the oligotrophic ocean, thus accurate estimates of $N_2$ fixation are of primary importance for oceanographers
to constrain and predict the evolution of marine biogeochemical carbon and N cycles. Global $N_2$ fixation estimates
have increased dramatically over the past three decades (Luo et al., 2012). The results of this study show that some
poorly explored areas such as the WTSP provide unique conditions for diazotrophs to fix at high rates and may
contribute to revise upward the current $N_2$ fixation estimates for the Pacific Ocean. Further studies would be required
along the annual time scale to assess the seasonal variability of $N_2$ fixation in this region and perform accurate N
budgets. Nonetheless, such high $N_2$ fixation rates question whether or not these high N inputs can balance the N losses
in the ETSP. A recent study based on the N* (the excess of N relative to P) at the South Pacific scale (Fumenia et al.,
Submitted) reveals a strong positive N* anomaly (indicative of $N_2$ fixation) in the surface and thermocline waters of
the WTSP, which potentially influences the geochemical signature of the thermocline waters further east in the South
Pacific through the regional circulation. However, the WTSP is chronically undersampled, and a better description of
the mesoscale and general circulation would be necessary to assess how N sources and sinks are coupled at the South
Pacific scale.










**Author contribution**: S.B designed the experiments, S.B. M.C., H.B. and M.B. carried them out at sea. M.C., H.B.,
R.A.F., S.H.N. and O.G. analyzed the samples, M.C. and S.B. analyzed the data. S.B. prepared the manuscript with
contributions from all co-authors
**Acknowledgements**
This research is a contribution of the OUTPACE (Oligotrophy from Ultra-oligoTrophy PACific Experiment) project
(https://outpace.mio.univ-amu.fr/) funded by the Agence Nationale de la Recherche (grant ANR-14-CE01-0007-01),
the LEFE-CyBER program (CNRS-INSU), the Institut de Recherche pour le Développement (IRD), the GOPS
program (IRD) and the CNES (BC T23, ZBC 4500048836). The OUTPACE cruise
(http://dx.doi.org/10.17600/15000900) was managed by the MIO (OSU Institut Pytheas, AMU) from Marseilles
(France). The authors thank the crew of the R/V L'Atalante for outstanding shipboard operations. G. Rougier and M.
Picheral are warmly thanked for their efficient help in CTD rosette management and data processing, as well as C.
Schmechtig for the LEFE-CyBER database management. Aurelia Lozingot is acknowledged for the administrative
work. All data and metadata are available at the following web address: http://www.obs-
vlfr.fr/proof/php/outpace/outpace.php. M.B. was funded by the People Programme (Marie Skłodowska-Curie Actions)
of the European Union's Seventh Framework Programme (FP7/2007-2013) under REA grant agreement number
625185. The participation, nucleic acid sampling and analysis were funded to RAF by the Knut and Alice Wallenberg
Foundation. RAF also acknowledges the assistance by Dr. Lotta Berntzon and Andrea Caputo.



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



1    **Table 1**. $^{15}N/^{14}N$ ratio of suspended particulate nitrogen (average over the photic layer) across the OUTPACE transect.

| Station # | $^{15}N/^{14}N$ ratio-PN$_{susp}$ (‰) |
|---|---|
| *MA waters* | |
| 1 | 2.00 |
| 2 | 0.78 |
| 3 | 0.57 |
| A | - |
| 4 | 2.71 |
| 5 | 1.57 |
| 6 | 1.91 |
| 7 | 0.5 |
| 8 | -2.45 |
| 9 | -2.21 |
| 10 | -2.7 |
| 11 | -7.05 |
| 12 | 1.89 |
| B | -2.88 |
| *Average MA waters* | *-0.41* |
| *GY waters* | |
| C | 7.91 |
| 14 | 8.72 |
| 15 | 7.55 |
| *Average GY waters* | *8.06* |





1   **Table 2**. Summary of relationships between $N_2$ fixation and physical and biogeochemical parameters and among $N_2$

2   fixation rates and several diazotrophic or non-diazotrophic planktonic groups indicated by Spearman's rank correlation

3   (n=102, $\alpha$=0.05). The significant correlations (p<0.05) are indicated by an asterisk (*).

| | Variable | Unit | $N_2$ fixation<br>Spearman's correlation coefficient |
|---|---|---|---|
| *Physical and biogeochemical parameters* | Pressure | dbar | -0.705* |
| | Temperature | °C | 0.658* |
| | Salinity | psu | -0.701* |
| | Oxygen | $\mu mol\ Kg^{-1}$ | 0,151 |
| | PAR | $\mu mol\ photons\ m^{-2}\ s^{-1}$ | 0.319* |
| | $NO_3^-$ | $\mu mol\ L^{-1}$ | -0.544* |
| | $NH_4^+$ | $\mu mol\ L^{-1}$ | -0,024 |
| | DIP | $\mu mol\ L^{-1}$ | -0.770* |
| | $Si(OH)_4$ | $\mu mol\ L^{-1}$ | -0.724* |
| | DFe | $nmol\ L^{-1}$ | 0.398* |
| | DON | $\mu mol\ L^{-1}$ | 0.517* |
| | DOP | $\mu mol\ L^{-1}$ | 0.418* |
| | DOC | $\mu mol\ L^{-1}$ | 0.573* |
| | PON | $\mu mol\ L^{-1}$ | 0.721* |
| | POC | $\mu mol\ L^{-1}$ | 0.723* |
| | Biogenic silica | $\mu mol\ L^{-1}$ | 0.274* |
| | Chl *a* | $\mu g\ L^{-1}$ | 0.266* |
| | Primary production | $\mu g\ C\ L^{-1}\ d^{-1}$ | 0.657* |
| | Bacterial production | $\mu mol\ C\ L^{-1}\ h^{-1}$ | 0.692* |
| | $T_{DIP}$ | days | -0.721* |
| *Planktonic groups* | *Trichodesmium* sp. | *nifH* copies $L^{-1}$ | 0.729* |
| | UCYN-A1 | *nifH* copies $L^{-2}$ | -0,051 |
| | UCYN-A2 | *nifH* copies $L^{-3}$ | -0,147 |
| | UCYN-B | *nifH* copies $L^{-4}$ | 0.511* |
| | het-1 | *nifH* copies $L^{-5}$ | 0.538* |
| | het-2 | *nifH* copies $L^{-6}$ | 0.576* |
| | het-3 | *nifH* copies $L^{-7}$ | 0.276* |
| | *Prochlorococcus* sp. | cells $ml^{-1}$ | 0.697* |
| | *Synechococcus* sp. | cells $ml^{-1}$ | 0.720* |
| | Pico-eukaryotes | cells $ml^{-1}$ | -0.450* |
| | Bacteria | cells $ml^{-1}$ | 0.780* |
| | Protists | cells $ml^{-1}$ | 0.680* |




1    **Table 3**. Summary of diazotroph abundances and nanoSIMS analyses at SD2, SD6 and LDB.

| Station # | *Trichodesmium* abundance (cells L$^{-1}$) | Contribution to diazotroph community (%) | Atom% $^{15}$N (mean $\pm$ SD) | N$_2$ fixation rate (fmol cell$^{-1}$ d$^{-1}$) | Contribution to bulk N$_2$ fixation (%) |
|---|---|---|---|---|---|
| SD2 | $1.3 \times 10^5$ | 84.9 | $0.610 \pm 0.269$ | $38.9 \pm 8.1$ | 83.8 |
| SD6 | $3.3 \times 10^5$ | 68.0 | $0.637 \pm 0.355$ | $29.3 \pm 5.4$ | 47.1 |
| LDB | $1.2 \times 10^5$ | 91.8 | $0.981 \pm 0.466$ | $123.8 \pm 24.7$ | 52.9 |

| Station # | *Trichodesmium* abundance (cells L$^{-1}$) | Contribution to diazotroph community (%) | Atom% $^{15}$N (mean $\pm$ SD) | N$_2$ fixation rate (fmol cell$^{-1}$ d$^{-1}$) | Contribution to bulk N$_2$ fixation (%) |
|---|---|---|---|---|---|
| SD2 | $2.0 \times 10^4$ | 13.2 | $1.163 \pm 0.531$ | $30.0 \pm 6.4$ | 10.1 |
| SD6 | $1.5 \times 10^5$ | 31.7 | $0.517 \pm 0.237$ | $6.1 \pm 1.2$ | 6.1 |
| LDB | $3.8 \times 10^2$ | 0.3 | n.d | n.d | n.d |





1    **Table 4**. Summary of average *nifH* gene expression data determined by qRT-PCR at selected stations (SD2, SD6,

2    LDB), where the cell-specific $N_2$ fixation rates were measured.

| Diazotroph | Station SD2 cDNA *nifH* (gene copies L$^{-1}$) | Station SD6 cDNA *nifH* (gene copies L$^{-1}$) | Station LDB cDNA *nifH* (gene copies L$^{-1}$) |
|---|---|---|---|
| *Trichodesmium* spp. | $1.1 \times 10^5$ | $5.1 \times 10^5$ | $5.78 \times 10^4$ |
| UCYN-B | $1.9 \times 10^5$ | $1.5 \times 10^5$ | $1.03 \times 10^2$ |
| Het-1 | $6.83 \times 10^2$ | $1.56 \times 10^3$ | $2.04 \times 10^2$ |
| Het-2 | $5.44 \times 10^2$ | $2.14 \times 10^2$ | bd |
| UCYN-A1 | bd | bd | bd |

31





**Figure captions**
**Figure 1.** Upper panel: general situation of the western and central Pacific and associated Seas. Lower panel: Sampling
locations superimposed on a composite sea surface Chl *a* concentrations during the OUTPACE cruise (February 19[th]-
April 3[rd], quasi-Lagrangian weighted mean Chl *a*). Short- duration (X) and long (+) duration stations are indicated.
The satellite data are weighted in time by each pixel's distance from the ship's average daily position for the entire
cruise. The white line shows the vessel route (data from the hull-mounted ADCP positioning system). The satellite
products are provided by CLS with the support of CNES.
**Figure 2.** Horizontal and vertical distributions of (a) seawater temperature (°C), (b) fluorescence ($\mu$g L$^{-1}$), (c) NO$_3^-$
($\mu$mol L$^{-1}$), (d) DIP ($\mu$mol L$^{-1}$) and (e) N$_2$ fixation rates (nmol N L$^{-1}$ d$^{-1}$) across the OUTPACE transect. LD stations
are reported as well as the two sub-regions MA: Melanesian archipelago waters, GY: South Pacific Gyre waters. Y
axis: pressure (dbar), X axis: longitude, black dots correspond to sampling depths.
**Figure 3**. (a) $^{15}$N/$^{14}$N ratio of the N$_2$ pool in the incubation bottles incubated either in on-deck incubators (n=54) and
in situ (mooring line) (n=44). The dashed line represent the theoretical value (~8.2 atom%) calculated assuming
complete isotopic equilibration between the gas bubble and the seawater based on gas constants. (b) Depth profiles of
$^{15}$N/$^{14}$N ratio of the N$_2$ pool in the incubation bottles incubated either in on-deck incubators (filled symbols) or on an
in situ mooring line (open symbols).










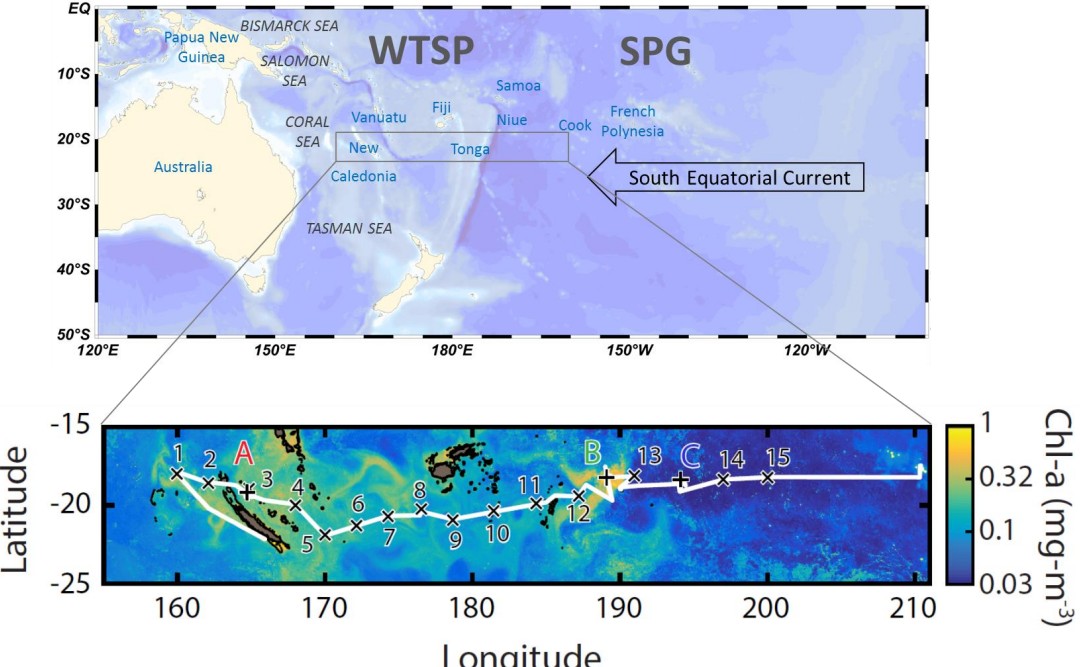

6                                Figure 1.



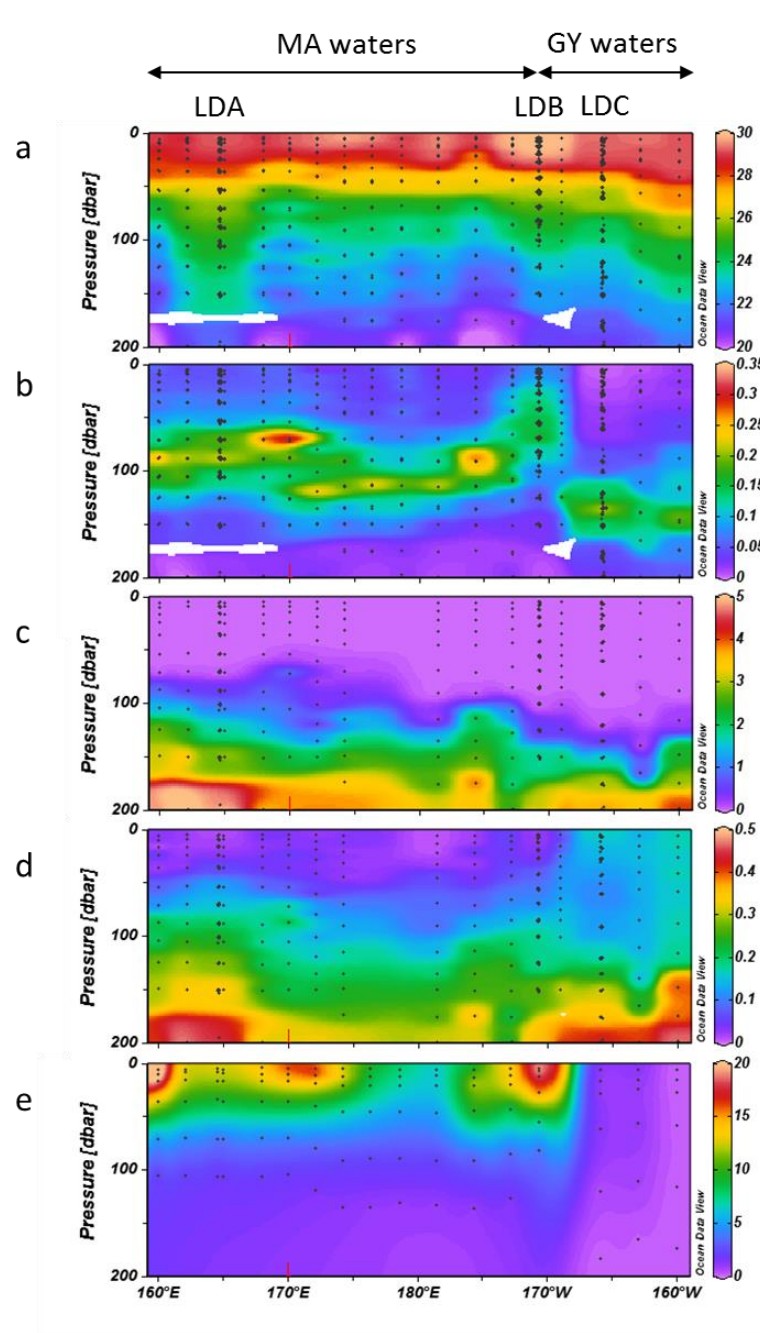

3         Figure 2.



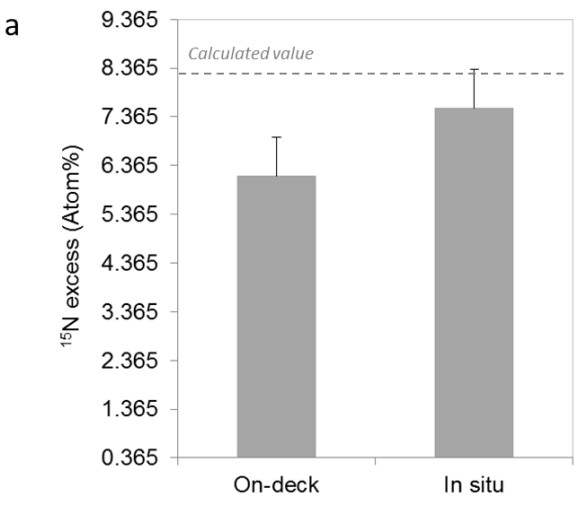

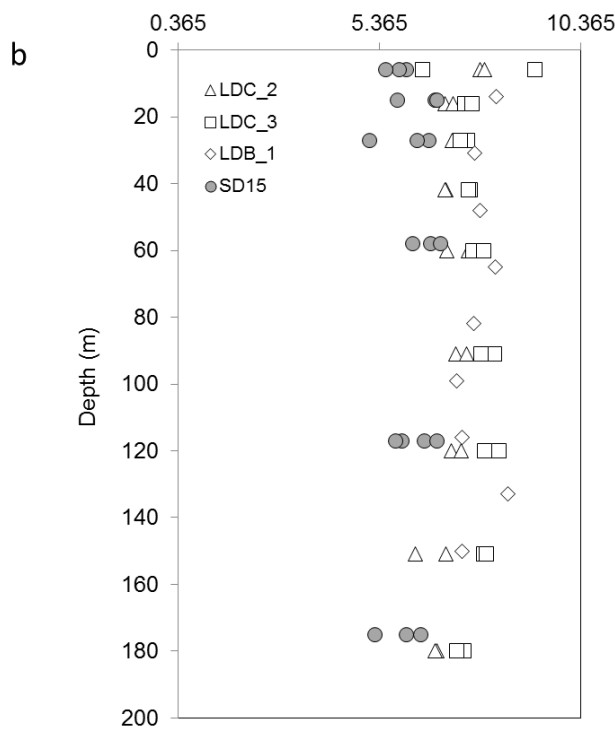

2                            Figure 3