# Peer review of "western tropical South Pacific hot spot of N2 fixation (OUTPACE"

_Biogeosciences, 2017_

## Referee Comment (RC1) · Anonymous Referee #1 · 25 Jan 2018

Review of "In depth characterization of diazotroph activity across the Western Tropical South Pacific hot spot of N2 fixation" by Bonnet et al.

Bonnet and coauthors measured nitrogen fixation rates and diazotroph abundance along a west-east transect in the western tropical South Pacific Ocean. They report some astonishingly high rates along this transect and offer explanations for the driving factors. This is a solid piece of work with some important and interesting findings. Most of my comments below are minor although there are a couple of major typographical errors that need to be fixed. But this manuscript can be improved to make it something more than a data report by providing a good oceanographic context to the observations

reported. It seems very likely that the researchers encountered different water masses along the transect that account for some of the variances in nitrogen fixation rates reported and providing that context would be useful. I suggest not using acronyms when not necessary – the difference between GY and gyre is two letters and it just makes it easier to read. I also suggest some minor modifications to the figures to make them more useful.

I do have a couple a couple of pet peeves to express and hope the authors will pay attention to at least the second and change the manuscript accordingly.

1) While I realize that this was a major oceanographic expedition with many groups, all working at different pace and thus necessarily, some results are available earlier and already published while others more recently processed, it is still frustrating to read to read a manuscript where critical bits of information are presented elsewhere, either already published, in review or in preparation. It is unfortunate that success in the modern scientific enterprise is measured by numbers of papers and careers of especially young scientists are determined by first authorships, resulting in piece meal papers. I don't expect the authors can do much about this but do wish to raise this issue because it is especially important for major interdisciplinary field expeditions such as OUTPACE.

2) The word "hotspot" is starting to get overused. It would seem that each investigator's favorite geography is a "hotspot" and I am having a difficult time with the concept of claiming a quarter or even one eighth of the largest ocean (western tropical South Pacific) as a hotspot. As the authors themselves say, "WTSP is a vast oceanic region" (page 2, line 21). The data to support the idea that the entire WTSP is a hotspot is still sparse and much too variable - 631286 in Melanesian archipelago waters - is a range of almost 45% in this cruise alone. The findings in this manuscript are significant even without that claim. In addition, there is one, potentially two real hotspots within this transect that are important in my view and that get lost when the claim is made to the whole area – I am both supportive and excited by the idea that there is a "sweet spot"

(to use a different term) for diazotrophy at the interface where there is a supply of iron and phosphorus (around station LDB). This zonal gradient is similar to the meridional gradient in Fe and P that Moore et al characterized in the Equatorial Atlantic. But the South Pacific is more complex and thus interesting in that there is clearly some sort of island effect with higher rates closer to the islands as well as Fe supply from the seafloor.

Specific comments and suggestions:

Page 2, line 5 – is it ammonia or amino acids?

Page 7, line 4 – should be per cell, not par cell

Page 7, line 18 (and elsewhere) – it would be good to discuss what is special about station LDB. This station is clearly a hotspot. Why? Was there any eddy activity here? Why is the water warmer here? Why is the chl higher all along the water column? Are we at the edge of water masses? Actually for that matter, what is going on at LDA where warmer waters are mixed down to almost 150 m.

Page 8, line 2 – what is DL? If it is detection limit, what is it? Why is the range 0-4 in line 4 – i.e. why is this not from detection limit?

Page 9, lines 34-38. This is a very important and interesting finding. While pressure could be one reason, clearly temperature would also play a role (although that would be in the opposite direction?) – i.e. there is a temperature gradient of 6-8 C between surface and 150m.

Page 10, line 11 – suggest using "differing" or some other word such as changing rather than differed?

Page 10, line 15 – what does under in-situ-simulated mean? Why not just on-deck incubations that simulated appropriate light levels?

Page 10, line 21 – it would appear that there is quite a bit of variability even in the

archipelago waters. I am concerned that the contouring for figure 2 makes it appear as though it is a lot more uniform than it really is. While I understand the attraction of presenting the data along a linear transect this way, I do worry that real numbers are getting lost in this presentation and that the rates are actually a whole bunch more variable.

Page 11, lines 3-22 – why only discussion of DIP – what about DOP? Trichodesmium can use DOP and it would have been interesting to see what was going on with that.

Page 11, lines 26-27 – what is the range for the DFe concentrations?

Page 12, lines 15-16 – Is it not the other way around – PAR explains depth?

Page 12, line 35 – dominated not dominating

Page 13, line 28 – suggest saying "more than" rather than above

Page 14, line 18-19 – the sentence construction suggests that rates have gone up rather than our understand of rates have changed

Page 24, Table 3. It would seem that the table header for the second row is wrong. Spent a lot of time trying to figure out why the numbers were different till I figured out that it is actually for UCYN B rather than Trichodesmium

Page 25, Table 4. Why are the numbers for cDNA gene copies different from that reported in the text?

Page 27 Figure 1: Suggest improving 1a and show the ocean currents better – the superimposition of a big arrow does not do much. I am not clear how 1b was done – am just surprised that there are no clouds in the image. This is not critical expect to understand if the high chl patches seen are temporally relevant.

Page 28, Figure 2: Suggest adding the parameters to the various subfigures.

---

## Referee Comment (RC2) · C. Löscher (Referee) · 1 Feb 2018

Bonnet and colleagues fill again one of the white spots on the maps of nitrogen fixation by presenting data from the OUTPACE cruise in 2015 to the western and central tropical South Pacific. N2 fixation rates from the photic are resented and complemented with a metadata analysis and a mining for the major diazotrophic clades Trichodesmium and UCYN-B. N2 fixation can be massive in that region, which is shown to be split in two different biogeochemical parts, and can reach up to 631 $\pm$ 286 $\mu$mol N/m/d. Those rates are extremely high and thus not only confirm the predicted importance of that region for N2 fixation but actually show that the N supply via N2 fixation is much higher

than predicted. Moreover, they challenged the statistical approach of Luo and colleagues suggesting SST as a key regulating factor of N2 fixation in that region. This study now provides some realistic evidence for a limitation by either DIP or Fe. A discussion on the contribution of N2 fixation to primary and export production again emphasizes the importance of this region, which is from my point of view the best part of the manuscript. As in previous studies, I am missing direct C fixation rates, this would be easy to do in future studies and make your point of N2 fixation to primary production much stronger. Altogether, this is another piece of high-quality work of this group on this critical topic. The presentation is clear, the figures and tables are well chosen. I don't quite get, why there is no abstract, maybe it would be good to directly summarize your key findings, there. I only have minor comments and recommend publication in Biogeosciences.

Specific comments:

p. 2, l. 3 the biological carbon pump: the sentenced is phrase in a way that it sounds strange, here, also it deserves the Azam reference. Given your discussion on the contribution of N2 fixation to the carbon pump, this needs more than a sentence, please explain the connection between N2 fixation, C fixation and export production in some sentences.

p. 2, l. 6. nifH is only one gene, and it codes for one subunit of the nitrogenase reductase

p.4, l. 3 spell out $HgCl_2$

p.4, l. 8 and is throughout the text: submitted is capitalized, please change

p.4, l. 10 As the paper is in review a brief description of the method would be helpful

p.4, l. 13 ff and discussion: I see and understand why you chose for the bubble method here and I respect that you give me an argument to not start this discussion, again. I assume my opinion on the method issue is anyway clear by now, and seeing your

discussion shows how reflective and aware of the importance of that topic you are . For the sake of having one day a truly comparative method, let me add a very pragmatic argument: I used both methods excessively, as a practical conclusion I can conclude (and I am sure you know what I am talking about) shaking the bottles with the gas bubbles in them for 100 times each is much more exhausting then pre-dissolving the gas in a bag with water. This limits the number of bottles you can handle, and with that the spatial resolution you will get. In addition, if you try shaking on a culture of Trichodesmium, you kill parts of the culture as they seem to dislike the shaking even more than I do. This would lead to a possible underestimation in addition to the one caused by the method. I appreciate the text you added to the discussion, but I feel it is a bit of an overkill. I would recommend to shorten it, but leave the justification with the DOM and Fe contamination potential in, so that everyone can understand why you chose for it. In addition, Grosskopf et al showed, and this is actually what plays into your cards, that the bubble method is mostly problematic for low rates- you have massive rates, so you don't really run into that problem.

p. 5, l.5: Could you add a sentence on what a trapezoidal integration is?

p. 6, l.2 ff You follow your classical screening approach, here, and there is nothing bad about it. Still, I would be extremely curious seeing a nifH sequencing doe on those samples- how else can we know who is doing the N2 fixation, who could do it if conditions change? In the results it seems, your single cell rates add up to the bulk at station SD2 (I am referring to table 3), not at the others- so there must be something else. In this context, I appreciate the gene expression assay and indeed it could be that the het groups fill the gap. But obviously, the microbes you don't look at will not show up. I see, you are referring to Stenegren et al., 2017, but it would be beneficial for future studies to have a proper deep sequencing included. Bonnet and colleagues fill again one of the white spots on the maps of nitrogen fixation by presenting data from the OUTPACE cruise in 2015 to the western and central tropical South Pacific. N2 fixation rates from the photic are resented and complemented with a metadata

analysis and a mining for the major diazotrophic clades Trichodesmium and UCYN-B. N2 fixation can be massive in that region, which is shown to be split in two different biogeochemical parts, and can reach up to 631 $\pm$ 286 $\mu$mol N/m/d. Those rates are extremely high and thus not only confirm the predicted importance of that region for N2 fixation but actually show that the N supply via N2 fixation is much higher than predicted. Moreover, they challenged the statistical approach of Luo and colleagues suggesting SST as a key regulating factor of N2 fixation in that region. This study now provides some realistic evidence for a limitation by either DIP or Fe. A discussion on the contribution of N2 fixation to primary and export production again emphasizes the importance of this region, which is from my point of view the best part of the manuscript. As in previous studies, I am missing direct C fixation rates, this would be easy to do in future studies and make your point of N2 fixation to primary production much stronger. Altogether, this is another piece of high-quality work of this group on this critical topic. The presentation is clear, the figures and tables are well chosen. I don't quite get, why there is no abstract, maybe it would be good to directly summarize your key findings, there. I only have minor comments and recommend publication in Biogeosciences.

Specific comments:

p. 2, l. 3 the biological carbon pump: the sentenced is phrase in a way that it sounds strange, here, also it deserves the Azam reference. Given your discussion on the contribution of N2 fixation to the carbon pump, this needs more than a sentence, please explain the connection between N2 fixation, C fixation and export production in some sentences.

p. 2, l. 6. nifH is only one gene, and it codes for one subunit of the nitrogenase reductase

p.4, l. 3 spell out HgCl2

p.4, l. 8 and is throughout the text: submitted is capitalized, please change

p.4, l. 10 As the paper is in review a brief description of the method would be helpful

p.4, l. 13 ff and discussion: I see and understand why you chose for the bubble method here and I respect that you give me an argument to not start this discussion, again. I assume my opinion on the method issue is anyway clear by now, and seeing your discussion shows how reflective and aware of the importance of that topic you are . For the sake of having one day a truly comparative method, let me add a very pragmatic argument: I used both methods excessively, as a practical conclusion I can conclude (and I am sure you know what I am talking about) shaking the bottles with the gas bubbles in them for 100 times each is much more exhausting then pre-dissolving the gas in a bag with water. This limits the number of bottles you can handle, and with that the spatial resolution you will get. In addition, if you try shaking on a culture of Trichodesmium, you kill parts of the culture as they seem to dislike the shaking even more than I do. This would lead to a possible underestimation in addition to the one caused by the method. I appreciate the text you added to the discussion, but I feel it is a bit of an overkill. I would recommend to shorten it, but leave the justification with the DOM and Fe contamination potential in, so that everyone can understand why you chose for it. In addition, Grosskopf et al showed, and this is actually what plays into your cards, that the bubble method is mostly problematic for low rates- you have massive rates, so you don't really run into that problem.

p. 5, l.5: Could you add a sentence on what a trapezoidal integration is?

p. 6, l.2 ff You follow your classical screening approach, here, and there is nothing bad about it. Still, I would be extremely curious seeing a nifH sequencing doe on those samples- how else can we know who is doing the N2 fixation, who could do it if conditions change? In the results it seems, your single cell rates add up to the bulk at station SD2 (I am referring to table 3), not at the others- so there must be something else. In this context, I appreciate the gene expression assay and indeed it could be that the het groups fill the gap. But obviously, the microbes you don't look at will not show up. I see, you are referring to Stenegren et al., 2017, but it would be beneficial
for future studies to have a proper deep sequencing included.

---

## Referee Comment (RC3) · Anonymous Referee #3 · 8 Feb 2018

The manuscript presented by Bonnet et al. reports N2 fixation rate measurements and diazotroph abundances from the Western tropical South Pacific. Complementary single-cell measurements of the two most abundant diazotrophs reveal the biogeochemical importance of each of these organisms in this region. Since this manuscript reports a subset of data collected on the OUTPACE cruise, these measurements are analyzed in correlation to a comprehensive set of nutrients, including dissolved iron, and other biogeochemical parameters. The manuscript is well written and the conclusions of this manuscript are appropriately based on the presented data. This new data is a nice addition of N2 fixation rate measurements in relation to biogeochemical parameters and a contribution of individual organisms (not solely based on abundances)

[Figure]

that will ultimately help confine the extent and magnitude of N input by N2 fixation into the global ocean. I only have a few comments (please see below).

Abstract: Maybe the authors could add a little more discussion/conclusions to the abstract as it currently reads almost like results only.

p 3, l 5: I would use either ammonia (NH3) or ammonium (NH4+) p 3, l 6: Isn't this 'nif genes' rather than 'nifH genes'? p 3, l 7: anammox uses nitrite and ammonium as substrates, maybe those could be added to the fixed N? Section 2.2: As far as I understand, the authors used the time-zero samples rather than incubated controls as the natural abundance value in the N2 fixation rate calculation. In many cases, this is OK; however, I have also seen quite large changes in the natural abundance over time in incubated samples/bottles without the addition of stable isotope. These are usually the result of fractionation during the incubation time, e.g. due to the uptake of residual nitrate or remineralization of organic material. The fractionation effects can lead to higher or lower d15N values of the natural abundances. In the absence of incubated controls, the detection limits of N2 fixation might be a bit worse than if those values are available. I would therefore recommend that the authors add their detection limits to the manuscripts, such as the minimum change in d15N that was used as a cutoff for a significant 15N enrichments or reporting the actual d15N values measured in their incubations and the time-zero values. This would also be coherent with general criticisms brought up in the recent paper by Gradoville et al. (2017; DOI: 10.1002/lno.10542).

Section 2.4: Was the at% 15N in the N2 pool measured here as well?

Section 3.4.: Was primary production measured on this cruise? Based on the description in this section, it sounds to me as if N2 fixation somewhat scales with productivity and/or turnover of organic material.

p 15, l 5: I assume that the "%" dropped from the 9.7? If not, does that mean that more organic matter is exported than produced at a given point in time?

p 15, l 14: With respect to the structure of the sections, I would almost move the entire section 4.2 here, as the rest of the discussion nicely scales from a more detailed and organism-centric discussion to a more system-oriented discussion. This would also have the side effect that the end of your discussion is not quite as focused on so many references that are related to other OUTPACE data which are not actually presented here.

Table 1: Any idea on why the d15N value is so low for stn 11 (i.e. -7.05 ‰? Do you have any depth distribution of the d15N values for PN?

---

## Referee Comment (RC4) · Anonymous Referee #4 · 10 Feb 2018

Review of the manuscript: In depth characterization of diazotrophs activity across the Western Tropical South Pacific hot spot of N2 fixation by S. Bonnet et al. I can't recommend this manuscript for publication in its current form, which is a shame, because it partially presents a very interesting data set about a large, but relatively understudied part of the world ocean. I have made this decision primarily because the authors do not present sufficient data or a convincing analysis to make their case. I think they need to show the reader a lot more information, a more expansive and rigorous analysis and be more up front about how they arrive at their conclusions. At present, I am unconvinced. I don't think that the problem can be simply rectified by rejigging the existing manuscript, adding some additional figures/tables or doing more statistics as in

a normal 'major' revision. The authors need to make a fresh start, think carefully about where they want to go and the strengths/limitations of their data, and then re-write carefully. If they do that properly, I think this work has the potential to make a very significant contribution to the global N-fixation literature. Some general comments: This reviewer recognizes that English may not be the lead author's prime language. However, in places, some of the word choices and syntax are ambiguous so that different readers are likely to take different things away from the manuscript. Specific examples are noted below. The authors should take advantage of the native English speakers in the author list to ensure that the wording is done with more precision to ensure that the intended meaning is communicated. Repeated references are made throughout to works which are 'submitted' or 'in review'. While this indicates the present manuscript is quite timely, the reader can make no critical or objective use of these references as they haven't seen the light of day, might be rejected or heavily altered before they are eventually published. It's hard to take these citations at face value. Where possible, I would include actual data from closely related studies in your paper to genuinely demonstrate your point, noting by citation that a fuller description will be published in other work. If not possible, stick to your own dataset. The authors need to be a lot more precise about your geography. The 'regions' used herein are quite large, loosely defined and contain a number of somewhat similar, but different oceanographic regimes. For example, 'Western Tropical South Pacific' would also include the Warm Pool region north of PNG – it's western, south of the equator and certainly tropical – but a different setting altogether. Likewise, 'Eastern Tropical South Pacific' also includes extensive areas of Ekman driven upwelling where it would be difficult to extrapolate your measurements. The longitude scale on Fig. 1 (bottom) is seriously wrong. For starters, what exactly do you mean by a N-fixation 'hot spot'? What's the cut-off? A summary table at least summarizing ranges of measured or reliably estimated N-fixation in other 'hot-spot' parts of the world ocean (e.g. Arabian Sea, Caribbean, Arafura Sea, etc., etc.) would be very useful to set the scene and would tie your work into the wider literature of global N-fixation. As a reader, I'm thinking a lot about how is this paper

compares with the much larger body of published work done by Capone, Carpenter and their collaborators/students, etc. There seems to be little quantitative integration (show me the numbers) with even the many more recent measurements of N-fixation in the SW Pacific. These need to be tied together, or shown why not. Figures and tables – Need more of them!, and more quantitative! ODV color contour plots are nice, but awfully hard to interpret quantitatively, and quite impossible if looking at a B/W version of your paper. Show some convincing/representative profiles, hard contour lines and East-West quantitative values. The discussion, by and large, is mostly hand-waving and speculation. While quite a few papers are cited (several of which haven't been published), there is a lack of quantitative information and data presented from these studies with which to compare the authors' results, assess their veracity and draw comparisons. The bit about regional differences being due to iron (etc.) inputs from submarine volcanos is wholly speculative on the information provided. Not a shred of quantitative information is presented to back this assertion up. Might the regional difference in fixation be due to regional differences in wind stress and water masses which affects the depth of the mixed layer and vertical mixing through the thermocline? I'd like to see a more focused discussion. Why no comparison with the extensive work done on N-fixation, fluxes and driving processes done at station ALOHA? An opportunity is missed.

Some specific comments: Page Line(s) Comment 1 2 What exactly do you mean by 'hot spot' – see above 3 10-11 By 'highest rates of N2 fixation' do you mean on an area-specific basis (I doubt it) or aggregate fixation on a regional basis primarily because of the very large area involved? The oxygen deficient zones of the eastern Pacific are due to higher regional productivity arising from Ekman-driven upwelling at the basin scale, not N-fixation. Indeed, fixation tends to be lower in upwelling regions. Simple N:P ratios are a poor predictor in this case. 10-20 This paragraph seems to do a logical U-turn 21 etc. References to regional N-fixation ignore large database of published historical estimates by Capone, Carpenter, etc., etc., etc. 4 11 Change "equalled to" to "reliably extrapolated to estimates of" 18 Change "previously undocumented" to "new"

22 What is the basis of 'selection' – suggest leave out 22 What do you mean by 'potential ecological impact' of N-fixation? Poor wording. 29 Suggest changing 'contrasted' to something about a gradient of conditions. What's the essential difference between 'oligotrophic' and 'ultra-oligotrophic'? Strictly speaking that's like saying something is 'more better'. 33 (etc.) By 'fluorescence', I presume you mean 'chlorophyll fluorescence' – say it because lots of other things fluoresce if you measure it right. 5 4 Use 'stored' instead of 'preserved. More importantly for low-level nutrient analyses – how long were the samples stored before analysis (hours, days, months)? 7-10 Are there any actually published papers that describe these methods? Preferable. In the case of dissolved iron, the more widely used and less confusing notation would be: Fediss. 14 In using open-ocean communities, it is almost universally observed that metal and organic contamination results in under-estimation of rates due to toxicity of these materials to finicky oceanic bugs. Why do you think they are over-estimated? 20-21 Presumably you mean sub-samples taken from the Niskins. Explicit reading suggests you collected the water in situ in the poly-carb bottles. 33 How were these filters stored and for how long? Text suggests they were analysed almost right-away (good if you can do it correctly!). 6 20 What's the "them"? 26-28 What are the flow cytometric characteristics you sorted and counted the UCYN cells with? A lot of those don't have much, if any photosynthetic pigment, and if they did, the near-surface ones would likely be bleached a bit? It would be nice to see a cytogram. 8 2 Strictly speaking, you're 'estimating' this, not determining it. 16 (etc) By 0-50 m, I presume you mean the 'surface mixed layer'. This is a key matter herein as the surface mixed layer thickness changes along your transect. Strictly speaking, surface is surface (say 0-5 m). Even small and ephemeral density gradients in the near-surface layer and surface mixed layer can have profound influences on vertical mixing rates and hence the light histories of cells embedded in the surface layer. Be very specific! 23 By DCM, I presume you mean 'deep chlorophyll maximum'? 29-34 It's not clear what, if anything, this paragraph contributes to the paper. 37-38 This seems very wrong. I'm presuming you actually mean the per-mil deviation of the particulate matter ($\delta$15N) from the normal natural abundance of 15N

(0.367%). Normally, N-fixation has a $\delta$15N value close to 0‰. Larger deviations would suggest other fractionation processes such as denitrification. Clarify and fix up. 9 1-8 This is all the 'new' N-fixation data in the results text. Pretty thin. As an interested party, I'd like to see a lot more. Graphics and tables too. 10-17 Correlations – So what? Tables of correlation coefficients fill space, but are instantly forgettable. What's the point of the correlations other than that you can do them? 18-23 Show 'em or ditch this. 10 1-5 etc Decimals on figures. It's easy to calculate lots of decimal places on figures, but they clutter up the text. Given the analytical and natural variability of these processes and the analytical processes, how many decimal places are really justified and meaningful? 17-18 Realistically, one never really 'measures' a process. Given all of the factors at work, the best we can do is 'estimate' its magnitude. Best to be frank about that. 10-11 16-9 You probably overdid this bit of text. The 'bubble' problem is well known. Best to just say that you used the Montoya method to minimize contamination, but corrected for incomplete dissolution by measuring the 15N/14N ratio. It is interesting that you get higher dissolution in the samples incubated in situ and that needs to be explicitly corrected for

28 Fig. 1 Bottom: longitude scale is very wrong. Would like to see some comparative profiles of measured variables in different regions 29 Fig. 2 ODV scales need to be properly annotated. Some hard contours would be very useful 30 Fig. 3 Potentially useful, but. . ... Bottom: should have x-axis scale 0-10 with vertical dotted lines clearly showing natural abundance of 15N (0.367%) and the theoretical 15N/14N ratio if all 15N2 in bubble dissolved. Is the (very) slight mid-water increase in 15N excess statistically valid?

---

## Author Comment (AC1) · 2 May 2018

Response to Referee #1

We thank Reviewer #1 for his constructive comments. Below are copied the comments in regular font with our point by point responses below. Changes in the manuscript appear in 'track change' mode.

Bonnet and coauthors measured nitrogen fixation rates and diazotroph abundance along a west-east transect in the western tropical South Pacific Ocean. They report some astonishingly high rates along this transect and offer explanations for the driving

factors. This is a solid piece of work with some important and interesting findings. Most of my comments below are minor although there are a couple of major typographical errors that need to be fixed. But this manuscript can be improved to make it something more than a data report by providing a good oceanographic context to the observations reported. It seems very likely that the researchers encountered different water masses along the transect that account for some of the variances in nitrogen fixation rates reported and providing that context would be useful. I suggest not using acronyms when not necessary – the difference between GY and gyre is two letters and it just makes it easier to read. I also suggest some minor modifications to the figures to make them more useful.

I do have a couple of pet peeves to express and hope the authors will pay attention to at least the second and change the manuscript accordingly. 1) While I realize that this was a major oceanographic expedition with many groups, all working at different pace and thus necessarily, some results are available earlier and already published while others more recently processed, it is still frustrating to read a manuscript where critical bits of information are presented elsewhere, either already published, in review or in preparation. It is unfortunate that success in the modern scientific enterprise is measured by numbers of papers and careers of especially young scientists are determined by first authorships, resulting in piece meal papers. I don't expect the authors can do much about this but do wish to raise this issue because it is especially important for major interdisciplinary field expeditions such as OUTPACE.

I totally understand this comment and share this view in some ways. The OUTPACE special issue is divided in 27 papers, each dealing with a specific part of the 'story'. Since the submission of this manuscript, several of the cited papers are now available online on the special issue webpage https://www.biogeosciences.net/special_issue894.html, and some are accepted, which should improve accessibility. I acknowledge that a synthesis paper would be necessary to give a broad and multidisciplinary view of the ecosystem functioning in this region.
2) The word "hotspot" is starting to get overused. It would seem that each investigator's favorite geography is a "hotspot" and I am having a difficult time with the concept of claiming a quarter or even one eighth of the largest ocean (western tropical South Pacific) as a hotspot. As the authors themselves say, "WTSP is a vast oceanic region" (page 2, line 21). The data to support the idea that the entire WTSP is a hotspot is still sparse and much too variable - 631286 in Melanesian archipelago waters - is a range of almost 45% in this cruise alone. The findings in this manuscript are significant even without that claim. In addition, there is one, potentially two real hotspots within this transect that are important in my view and that get lost when the claim is made to the whole area – I am both supportive and excited by the idea that there is a "sweet spot" (to use a different term) for diazotrophy at the interface where there is a supply of iron and phosphorus (around station LDB). This zonal gradient is similar to the meridional gradient in Fe and P that Moore et al characterized in the Equatorial Atlantic. But the South Pacific is more complex and thus interesting in that there is clearly some sort of island effect with higher rates closer to the islands as well as Fe supply from the seafloor.

We agree with this comment and made a more reasonable use of the term hot spot throughout the new version of the manuscript. In addition, we now discuss the possible origin of the high N2 fixation rates at LDB, in relation with physical parameters and nutrient inputs page 15 line 10-28.

Specific comments and suggestions: Page 2, line 5 – is it ammonia or amino acids?

It is first transformed into ammonia (see reaction below) then in amino acids $N_2 + 8\ H^+ + 8\ e^- + 16ATP \rightarrow 2\ NH_3 + H_2 + 16\ ADP + 16\ Pi$

Page 7, line 4 – should be per cell, not par cell

Yes, sorry, this has been fixed

Page 7, line 18 (and elsewhere) – it would be good to discuss what is special about

station LDB. This station is clearly a hotspot. Why? Was there any eddy activity here? Why is the water warmer here? Why is the chl higher all along the water column? Are we at the edge of water masses? Actually for that matter, what is going on at LDA where warmer waters are mixed down to almost 150 m.

We added a section regarding the bloom at LDB page 15, line 23-28 and also refer to a video showing the evolution of the origin of the bloom: 'However, the huge surface bloom observed at LDB (Figure 1) and extensively studied by (de Verneil et al., 2017) was mainly sustained by N2 fixation (secondary fueling picoplankton and diatoms) (Caffin et al., in review, 2018)), rather than deep nutrient inputs (de Verneil et al., 2017). This bloom had been drifting eastwards for several months and initially originate from Fiji and Tonga archipelagoes (https://outpace.mio.univ-amu.fr/spip.php?article160), which may have provided sufficient Fe to alleviate limitation and triggered this exceptional diazotroph bloom as previously proposed by Shiozaki et al. (2014)'.

Page 8, line 2 – what is DL? If it is detection limit, what is it? Why is the range 0-4 in line 4 – i.e. why is this not from detection limit?

We acknowledge that DL was not defined in the submitted version. We have added in the Methods section the following sentence: 'The minimum quantifiable rate (quantification limit, QL) calculated using standard propagation of errors via the observed variability between replicate samples measured according to Gradoville et al. (2017) was 0.035 nmol N L-1 d-1.' We now refer to QL instead of DL in the Results section. Line 4 has been fixed as well.

Page 9, lines 34-38. This is a very important and interesting finding. While pressure could be one reason, clearly temperature would also play a role (although that would be in the opposite direction?) – i.e. there is a temperature gradient of 6-8 C between surface and 150m.

We acknowledge that there is temperature gradient with depth, but if temperature would
have played a role on dissolution, we would have had higher dissolution at depth (colder temperatures) compared to the surface, which was not observed. However, we thought that it was important to be mentioned and we have added a sentence as follows in the revised version: 'Despite the AN2 value was different according to the incubation mode, it did not change with the depth of incubation on the mooring line, indicating that a slightly higher pressure than atmospheric pressure (1.5 bar at 5 m depth) is enough to promote the 15N2 dissolution. It also indicates that the slightly lower seawater temperature (22-24°C) recorded at ∼100-180 m where the deepest samples were incubated likely did not affect the solubilization of the 15N2 gas'.

We partly agree with this comment. For samples collected above 50 m, the seawater temperature in the deck incubators and in situ was identical, so it seems that only pressure played a role in the higher dissolution. However, we acknowledge that seawater temperature is lower below 50 m and this may have enhanced the gas dissolution. Therefore, we have modified the text as follows: 'The seawater temperature checked regularly in the on-deck incubators was equivalent to ambient SST and likely did not explain the differences observed for samples collected above 50 m. However, we cannot exclude that the colder temperatures measured below 50 m-depth (∼23-26°C instead of 29-30°C in surface) may have, in addition to pressure, slightly enhanced the 15N2 gas dissolution, despite the AN2 value did not change with depth'.

Page 10, line 11 – suggest using "differing" or some other word such as changing rather than differed?

We changed by 'Despite the AN2 value was different according to the incubation mode'

Page 10, line 15 – what does under in-situ-simulated mean? Why not just on-deck incubations that simulated appropriate light levels?

We agree and changed the text accordingly

Page 10, line 21 – it would appear that there is quite a bit of variability even in the

archipelago waters. I am concerned that the contouring for figure 2 makes it appear as though it is a lot more uniform than it really is. While I understand the attraction of presenting the data along a linear transect this way, I do worry that real numbers are getting lost in this presentation and that the rates are actually a whole bunch more variable.

We now present all vertical profiles as supplementary information, clearly showing that rates are higher at some stations, which is also discussed in the new version of the manuscript.

Page 11, lines 3-22 – why only discussion of DIP – what about DOP? Trichodesmium can use DOP and it would have been interesting to see what was going on with that.

We have added a section regarding DOP in section 4.2: 'During the OUTPACE cruise, TDIPs were variable but were close or below two days in MA waters (Moutin et al., 2017), indicating a potential limitation by DIP at some stations. Trichodesmium, one of the major contributor to N2 fixation during the cruise, is known to synthesize hydrolytic enzymes in order to access the dissolved organic phosphorus pool (DOP) (Sohm and Capone, 2006). It is thus likely that DOP played a role in maintaining high Trichodesmium biomass in MA waters when TDIP locally dropped below two days as it was the case at station LD B for example (Moutin et al., 2017)'.

Page 11, lines 26-27 – what is the range for the DFe concentrations?

The range of DFe for MA waters was 0.2-66.2 nM and 0.2-0.6nM for GY waters. This has been added in the revised version page 14 lines 11-12.

Page 12, lines 15-16 – Is it not the other way around – PAR explains depth?

We changed the text as follows 'They were also negatively correlated with depth and logically positively correlated with PAR and seawater temperature, two parameters which are depth dependent'.

Page 12, line 35 – dominated not dominating

[Figure]

This has been fixed

Page 13, line 28 – suggest saying "more than" rather than above

I do not understand this comment

Page 14, line 18-19 – the sentence construction suggests that rates have gone up rather than our understand of rates have changed

Yes, we totally agree and have changed the sentence as follows: 'The number of N2 fixation estimates have increased dramatically at the global scale over the past three decades (Luo et al., 2012)'.

Page 24, Table 3. It would seem that the table header for the second row is wrong. Spent a lot of time trying to figure out why the numbers were different till I figured out that it is actually for UCYN B rather than Trichodesmium

Thanks a lot, it was a mistake, it is fixed in the new version.

Page 25, Table 4. Why are the numbers for cDNA gene copies different from that reported in the text?

I am not sure to get this comment. The cDNA numbers are not reported in the text. However, the nifH copies per liter from the qPCR assay are given in Table 3 and in the text to give the context of the nanoSIMS studies. Those numbers are different of course but it is well specified in the text page 10 lines 30-31 that we are not talking about cDNA in this section but about quantification of nifH.

Page 27 Figure 1: Suggest improving 1a and show the ocean currents better – the superimposition of a big arrow does not do much. I am not clear how 1b was done – am just surprised that there are no clouds in the image. This is not critical expect to understand if the high chl patches seen are temporally relevant.

The currents have been redrawn on Figure 1a. Figure 1b is actually a quasi-Lagrangian weighted mean Chl map in which the satellite data are weighted in time by each pixel's

distance from the ship's average daily position for the entire cruise. As there were a large number of images due to the 45 days duration of the cruise, the resolution is nice despite there were clouds.

Page 28, Figure 2: Suggest adding the parameters to the various subfigures.

The parameters have been added

---

## Author Comment (AC2) · 2 May 2018

Response to Referee #2

We thank Carolin Löscher for the time devoted to this review and for her constructive comments. Below are copied the comments in regular font with our point by point responses in blue. Changes in the manuscript appear in 'track change' mode.

Bonnet and colleagues fill again one of the white spots on the maps of nitrogen fixation by presenting data from the OUTPACE cruise in 2015 to the western and central tropical South Pacific. N2 fixation rates from the photic are resented and complemented

with a metadata analysis and a mining for the major diazotrophic clades Trichodesmium and UCYN-B. N2 fixation can be massive in that region, which is shown to be split in two different biogeochemical parts, and can reach up to $631\pm286\mu$mol N/m/d. Those rates are extremely high and thus not only confirm the predicted importance of that region for N2 fixation but actually show that the N supply via N2 fixation is much higher than predicted. Moreover, they challenged the statistical approach of Luo and colleagues suggesting SST as a key regulating factor of N2 fixation in that region. This study now provides some realistic evidence for a limitation by either DIP or Fe. A discussion on the contribution of N2 fixation to primary and export production again emphasizes the importance of this region, which is from my point of view the best part of the manuscript. As in previous studies, I am missing direct C fixation rates, this would be easy to do in future studies and make your point of N2 fixation to primary production much stronger. Altogether, this is another piece of high-quality work of this group on this critical topic. The presentation is clear, the figures and tables are well chosen. I don't quite get, why there is no abstract, maybe it would be good to directly summarize your key findings, there. I only have minor comments and recommend publication in Biogeosciences.

Specific comments:

p. 2, l. 3 the biological carbon pump: the sentenced is phrase in a way that it sounds strange, here, also it deserves the Azam reference. Given your discussion on the contribution of N2 fixation to the carbon pump, this needs more than a sentence, please explain the connection between N2 fixation, C fixation and export production in some sentences.

The first paragraph of the introduction has been modified according to the suggestions as follows: 'In the ocean, nitrogen (N) availability in surface waters controls primary production and the export of organic matter (Dugdale and Goering, 1967; Eppley and Peterson, 1979; Moore et al., 2013). The major external source of new N to the surface ocean is biological N2 fixation (100-150 Tg N yr-1, (Gruber, 2008)), the reduction of atmospheric di-nitrogen gas (N2) dissolved in seawater into ammonia (NH3+). The

process of N2 fixation is mediated by diazotrophic organisms that possess the nitrogenase enzyme, which is encoded by a suite of nif genes. These organisms provide new N to the surface ocean and act as "natural fertilizers", contributing to sustaining ocean productivity and eventually carbon (C) sequestration through the N2-primed prokaryotic C pump (Karl et al., 2003; Karl et al., 2012). This N source is continuously counteracted by N losses, mainly driven by denitrification and anammox, which convert reduced forms of N (nitrate, NO3-, nitrite NO2-, NH4+) into N2. Despite the critical importance of the N inventory in regulating primary production and export, the spatial distribution of N gains and losses in the ocean is still poorly resolved'.

p. 2, l. 6. nifH is only one gene, and it codes for one subunit of the nitrogenase reductase

We agree and the sentence has been changed as follows: 'The process of N2 fixation is mediated by diazotrophic organisms that possess the nitrogenase enzyme, which is encoded by a suite of nif genes'.

p.4, l. 3 spell out HgCl2

It has been spelled: 'fixed with mercuric chloride (HgCl2, final concentration 20 mg L-1)'

p.4, l. 8 and is throughout the text: submitted is capitalized, please change

We updated the references throughout the manuscript. Since the submission of this manuscript, most of the submitted ones are now under review or accepted. Anyways, when appropriate, we indicated 'submitted' without the capital S.

p.4, l. 10 As the paper is in review a brief description of the method would be helpful

The following paragraph has been added in section 2.1: 'Samples for determining dissolved Fe concentrations were collected and analyzed as described in Guieu et al. (Under review). Briefly, samples were collected using a Titane Rosette mounted with 24 teflon coated 12 L GoFlos deployed with a Kevlar cable. Dissolved Fe concentrations

were measured by flow injection with online preconcentration and chemiluminescence detection according to Bonnet and Guieu (2006). The reliability of the method was monitored by analyzing the D1 SAFe seawater standard (Johnson et al., 2007), and an internal acidified seawater standard was measured daily to monitor the stability of the analysis'.

p.4, l. 13 ff and discussion: I see and understand why you chose for the bubble method here and I respect that you give me an argument to not start this discussion, again. I assume my opinion on the method issue is anyway clear by now, and seeing your discussion shows how reflective and aware of the importance of that topic you are. For the sake of having one day a truly comparative method, let me add a very pragmatic argument: I used both methods excessively, as a practical conclusion I can conclude (and I am sure you know what I am talking about) shaking the bottles with the gas bubbles in them for 100 times each is much more exhausting then pre-dissolving the gas in a bag with water. This limits the number of bottles you can handle, and with that the spatial resolution you will get. In addition, if you try shaking on a culture of Trichodesmium, you kill parts of the culture as they seem to dislike the shaking even more than I do. This would lead to a possible underestimation in addition to the one caused by the method. I appreciate the text you added to the discussion, but I feel it is a bit of an overkill. I would recommend to shorten it, but leave the justification with the DOM and Fe contamination potential in, so that everyone can understand why you chose for it. In addition, Grosskopf et al showed, and this is actually what plays into your cards, that the bubble method is mostly problematic for low rates- you have massive rates, so you don't really run into that problem.

Yes I know what you are talking about ïĄŁ I will not give more details here as I can see that we understand each other. Ideally, we should have done proper comparisons on-board but as long as we have MIMS measurement, we can correct our data. Anyways, we shortened the discussion as suggested to keep only the 'contamination' argument, the rest is dedicated to the MIMS measurements during the in situ versus on-deck incubations. We also now make reference to a new study by Wannicke et al., (2018) and have added the following sentence: 'Moreover, a recent extensive meta-analysis (13 studies, 368 observations) between bubble and enriched amendment experiments to measure 15N2 rates reported that underestimation of N2 fixation is negligible in experiments that last 12-24 h (e.g. error is -0.2 %); hence our 24 h based experiments should be within a small amount of error (Wannicke et al., 2018)'.

p. 5, l.5: Could you add a sentence on what a trapezoidal integration is?

We have added the following sentence in the new version of the manuscript: 'Discrete N2 fixation rate measurements were depth-integrated over the photic layer using trapezoidal integration procedures. Briefly, the N2 fixation at each pair of depths is averaged, then multiplied by the difference between the two depths to get a total N2 fixation in that depth interval. These depth interval values are then summed over the entire depth range to get the integrated N2 fixation rate. The rate nearest the surface is assumed to be constant up to 0 m (JGOFS, 1988)'.

p. 6, l.2 If You follow your classical screening approach, here, and there is nothing bad about it. Still, I would be extremely curious seeing a nifH sequencing done on those samples- how else can we know who is doing the N2 fixation, who could do it if conditions change? In the results it seems, your single cell rates add up to the bulk at station SD2 (I am referring to table 3), not at the others- so there must be something else. In this context, I appreciate the gene expression assay and indeed it could be that the het groups fill the gap. But obviously, the microbes you don't look at will not show up. I see, you are referring to Stenegren et al., 2017, but it would be beneficial for future studies to have a proper deep sequencing included.

I totally agree with this comment. Indeed we cannot totally reconcile the single-cell rates of dominant groups with bulk rates, which mean that there are other players. Of course, DDAs quantified by Stenegren et al. are part of the story. We also observed some Katagnymene spiralis during the cruise, which despite scarce are really big and

may contribute to N2 fixation, together with the heterotrophs mentioned in Benavides et al. (same issue) and probably fungi etc.. so yes, deep sequencing is really needed to fill this gap. . .

---

## Author Comment (AC3) · 2 May 2018

Response to Referee #3

We thank reviewer 3 for the time devoted to this review and for his/her constructive comments. Below are copied the comments in regular font with our point by point responses in blue. Changes in the manuscript appear in 'track change' mode.

The manuscript presented by Bonnet et al. reports N2 fixation rate measurements and diazotroph abundances from the Western tropical South Pacific. Complementary single-cell measurements of the two most abundant diazotrophs reveal the biogeo-

chemical importance of each of these organisms in this region. Since this manuscript reports a subset of data collected on the OUTPACE cruise, these measurements are analyzed in correlation to a comprehensive set of nutrients, including dissolved iron, and other biogeochemical parameters. The manuscript is well written and the conclusions of this manuscript are appropriately based on the presented data. This new data is a nice addition of N2 fixation rate measurements in relation to biogeochemical parameters and a contribution of individual organisms (not solely based on abundances) hat will ultimately help confine the extent and magnitude of N input by N2 fixation into the global ocean. I only have a few comments (please see below).

Abstract: Maybe the authors could add a little more discussion/conclusions to the abstract as it currently reads almost like results only.

The abstract has been extended in the new version of the manuscript and now includes more discussion/conclusions.

p 3, l 5: I would use either ammonia (NH3) or ammonium (NH4+)

Totally right, this has been fixed

p 3, l 6: Isn't this 'nif genes' rather than 'nifH genes'?

Yes, the sentence has been changed as follows: 'The process of N2 fixation is mediated by diazotrophic organisms that possess the nitrogenase enzyme, which is encoded by a suite of nif genes'.

p 3, l 7: anammox uses nitrite and ammonium as substrates, maybe those could be added to the fixed N?

I had added 'anammox' just before submission thanks to a suggestion of a co-author, so yes of course, I have added these substrates in the new version of the manuscript.

Section 2.2: As far as I understand, the authors used the time-zero samples rather than incubated controls as the natural abundance value in the N2 fixation rate calculation.

In many cases, this is OK; however, I have also seen quite large changes in the natural abundance over time in incubated samples/bottles without the addition of stable isotope. These are usually the result of fractionation during the incubation time, e.g. due to the uptake of residual nitrate or remineralization of organic material. The fractionation effects can lead to higher or lower d15N values of the natural abundances. In the absence of incubated controls, the detection limits of N2 fixation might be a bit worse than if those values are available. I would therefore recommend that the authors add their detection limits to the manuscripts, such as the minimum change in d15N that was used as a cutoff for a significant 15N enrichments or reporting the actual d15N values measured in their incubations and the time-zero values. This would also be coherent with general criticisms brought up in the recent paper by Gradoville et al. (2017; DOI: 10.1002/lno.10542).

We acknowledge that isotopic fractionation may occur during the incubation period and that it would be generally more correct to use the natural abundance value after incubation. In the present case, the rates were so high so the impact of fractionation is probably negligible. However it may impact the quantification limit, and we have added a sentence regarding the minimum quantifiable rates in section 2.2: 'The minimum quantifiable rates (quantification limit, QL) calculated using standard propagation of errors via the observed variability between replicate samples measured according to Gradoville et al. (2017) were 0.035 nmol N L-1 d-1.

Section 2.4: Was the at% 15N in the N2 pool measured here as well?

It was measured in triplicates at every station but only in the bottles dedicated to bulk N2 fixation measurements and this value was used for the group-specific rate calculations as the same methodology was used (same bottles, same amount of 15N2 added) for both types of measurements.

Section 3.4.: Was primary production measured on this cruise? Based on the description in this section, it sounds to me as if N2 fixation somewhat scales with productivity

and/or turnover of organic material.

Yes PP was measured using the 14C labeling method (not 13C). It appears in the correlation table but not in text. We have thus added PP and bacterial production in this section (which are both correlated with N2 fixation). The sentence is now: 'Regarding the main biogeochemical stocks and fluxes measured during the cruise, N2 fixation rates were significantly positively correlated with dissolved Fe, dissolved organic N (DON), phosphorus (DOP) and carbon (DOC), particulate organic N (PON), particulate organic carbon (POC), biogenic silica (BSi), Chl a concentrations, primary and bacterial production (p<0.05), and significantly negatively correlated with NO3-, NH4+, DIP and silicate concentrations (p<0.05)'.

p 15, l 5: I assume that the "%" dropped from the 9.7? If not, does that mean that more organic matter is exported than produced at a given point in time?

Yes, this is 9.7 %, I have added the % in the new version of the manuscript.

p 15, l 14: With respect to the structure of the sections, I would almost move the entire section 4.2 here, as the rest of the discussion nicely scales from a more detailed and organism-centric discussion to a more system-oriented discussion. This would also have the side effect that the end of your discussion is not quite as focused on so many references that are related to other OUTPACE data which are not actually presented here.

I tried to do that and the structure of the manuscript did not seem coherent anymore to me. As the 4.2 section discusses the N2 fixation results, which are the first presented, it was not consistent for me to present detailed group-specific data before presenting the big picture. However, I acknowledge that section 4.4 Ecological relevance of N2 fixation in the WTSP contains many references of the OUTPACE SI not really related to the present study. Therefore, I propose to merge this section and the conclusion section to set our study in the general context of the OUTPACE study.

Table 1: Any idea on why the d15N value is so low for stn 11 (i.e. -7.05 ‰ Do you have any depth distribution of the d15N values for PN?

We measured d15N of PN at only 2 depths (the surface, 5 m and the deep chlorophyll maximum) so it's not a very good resolution. I was also wondering why such low values at SD 11. I know that this is the area where we have seen some high Fe inputs likely coming from underwater volcanoes. This may alter the isotopic composition of plankton. . .

---

## Author Comment (AC4) · 2 May 2018

The detailed reponses are on the Supplement attached document. The revised manuscript (in track-change mode) is also attached as a supplementary file.

Please also note the supplement to this comment: https://www.biogeosciences-discuss.net/bg-2017-567/bg-2017-567-AC4-supplement.pdf
* * *
[Figure]

**In depth characterization of diazotroph activity across the**

**western tropical South Pacific hot spot of N$_2$ fixation (OUTPACE**

**cruise)**

Sophie Bonnet[1,2], Mathieu Caffin[1], Hugo Berthelot[1], Olivier Grosso[1,] Mar Benavides[2,3], Sandra

Helias-Nunige[2], Cécile Guieu[4,5], Marcus Stenegren[6], Rachel A Foster[6]

[1]IRD, Aix Marseille Université, CNRS/INSU, Université de Toulon, Mediterranean Institute of Oceanography (MIO)

UM 110, 13288, Marseille-Nouméa, France-New Caledonia

[2]Mediterranean Institute of Oceanography (MIO) – IRD/CNRS/Aix-Marseille University IRD Nouméa, 101

Promenade R. Laroque, BPA5, 98848, Nouméa cedex, New Caledonia

[3]Marine Biology Section, Department of Biology, University of Copenhagen, 3000 Helsingør, Denmark

[4]Sorbonne Universités, UPMC Université Paris 06, CNRS, Laboratoire d'Océanographie de Villefranche (LOV),

06230 Villefranche-sur-Mer, France

[5]Center for Prototype Climate Modeling, New York University Abu Dhabi, P.O. Box 129188, Abu Dhabi, United Arab

Emirates

[6]Department of Ecology, Environment, and Plant Sciences, Stockholm University, Stockholm Sweden 10690

*Correspondence to*: Sophie Bonnet (sophie.bonnet@univ-amu.fr)

**Fig. 1.**

**Supplement:**

Response to Referee #4

Below are copied the comments in regular font with our point by point responses in blue.
Changes in the manuscript appear in 'track change' mode.

Review of the manuscript: In depth characterization of diazotrophs activity across the Western Tropical South Pacific hot spot of N2 fixation by S. Bonnet et al. I can't recommend this manuscript for publication in its current form, which is a shame, because it partially presents a very interesting data set about a large, but relatively understudied part of the world ocean. I have made this decision primarily because the authors do not present sufficient data or a convincing analysis to make their case. I think they need to show the reader a lot more information, a more expansive and rigorous analysis and be more up front about how they arrive at their conclusions. At present, I am unconvinced. I don't think that the problem can be simply rectified by rejigging the existing manuscript, adding some additional figures/tables or doing more statistics as in a normal 'major' revision. The authors need to make a fresh start, think carefully about where they want to go and the strengths/limitations of their data, and then re-write carefully. If they do that properly, I think this work has the potential to make a very significant contribution to the global N-fixation literature.

We are quite surprised that the reviewer opinion is that this paper do not present sufficient data. However, we believe that our story is compelling because:
-we provide accurate bulk N2 fixation over a 4000 km transect and the whole photic layer, which represent the first data for this region
-we tested several methods of incubations and provide a critic opinion of each method in the context of the methodological issues associated with N2 fixation measurements in the N2 fixation community
-we further investigate the contribution of the main diazotroph groups using single-cell isotope approaches (nanoSIMS)
-we provide extensive environmental data (dissolved iron, DIP turnover times…) to discuss the spatial variability of the N2 fixation activity observed. This part has been improved following the suggestions (please see below)

Some general comments: This reviewer recognizes that English may not be the lead author's prime language. However, in places, some of the word choices and syntax are ambiguous so that different readers are likely to take different things away from the manuscript. Specific examples are noted below. The authors should take advantage of the native English speakers in the author list to ensure that the wording is done with more precision to ensure that the intended meaning is communicated.

We acknowledge that English was not perfect as it is not the native language of the first author. The revised version of the manuscript has been checked by the native English speaker of the author list.

Repeated references are made throughout to works which are 'submitted' or 'in review'. While this indicates the present manuscript is quite timely, the reader can make no critical or objective use of these references as they haven't seen the light of day, might be rejected or heavily altered before they are eventually published. It's hard to take these citations at face value. Where possible, I would include actual data from closely related studies in your paper to genuinely demonstrate your point, noting by citation that a fuller description will be published in other work. If not possible, stick to your own dataset.

We totally understand this comment. The OUTPACE special issue is divided in 27 papers, each treating a specific part of the 'story'. Since the submission of this manuscript, several of the cited papers are now available online on the special issue webpage https://www.biogeosciences.net/special_issue894.html, and some are accepted (this has been updated in the revised version), which should improve the accessibility.

The authors need to be a lot more precise about your geography. The 'regions' used herein are quite large, loosely defined and contain a number of somewhat similar, but different oceanographic regimes. For example, 'Western Tropical South Pacific' would also include the Warm Pool region north of PNG – it's western, south of the equator and certainly tropical – but a different setting altogether. Likewise, 'Eastern Tropical South Pacific' also includes extensive areas of Ekman driven upwelling where it would be difficult to extrapolate your measurements. The longitude scale on Fig. 1 (bottom) is seriously wrong.

We agree that the WTSP is a vast region that includes the warm pool region north of PNG, but also the Solomon Sea, the Coral Sea etc... these regions were previously documented for N2 fixation, and we clearly state in the introduction section that our goal is to specifically study the central and eastern parts of the WTSP, that are critically undersampled + the border of the south pacific gyre. This was the goal of the OUTPACE project, so we only report results from this cruise. To avoid any misunderstanding, we have added 'OUTPACE cruise' in the title of the manuscript to specify that this article is about the results of this specific cruise and not the results about the entire WTSP.
The longitude scale on Figure here is the conventional one decided for all outpace papers of the Special issue. However we fixed that point in the revised version.

For starters, what exactly do you mean by a N-fixation 'hot spot'? What's the cut-off? A summary table at least summarizing ranges of measured or reliably estimated N-fixation in other 'hot-spot' parts of the world ocean (e.g. Arabian Sea, Caribbean, Arafura Sea, etc., etc.) would be very useful to set the scene and would tie your work into the wider literature of global N-fixation. As a reader, I'm thinking a lot about how is this paper compares with the much larger body of published work done by Capone, Carpenter and their collaborators/students, etc. There seems to be little quantitative integration (show me the numbers) with even the many more recent measurements of N-fixation in the SW Pacific. These need to be tied together, or shown why not.

To the best of our knowledge, there is no 'official' cut-off to design a hot spot but we clearly mentioned in the manuscript that the rates reported here are in the upper range of the higher caregory (100-1000 $\mu$mol N m$^{-2}$ d$^{-1}$) of rates reported in the N$_2$ fixation MAREDAT database for the global ocean (Luo et al., 2012). This was our argument to say that it is hot spot. However, as suggested by Reviewer 1, we tone down the hot spot story throughout the revised version. We also recently published the map below, which gathers N2 fixation rates from the Luo et al. global database (in green) compared to the rates measured in the WTSP (in red, this cruise and others) by our team. It speaks for itself and reveals the WTSP as a high N2 fixation area in the global ocean, at least from our current knowledge. We acknowledge that there are also other high N2 fixation areas such as the Caribbean Sea and probably other parts which are to date particularly undersampled. This map was published in a very short letter in PNAS to argue for a spatial decoupling between N2 fixation in the western Pacific and N losses in the eastern Pacific, and only present depth-integrated rates, with no details. The present paper aims at describing vertical and horizontal N2 fixation rates during the OUTPACE cruise in relation with environmental parameters, identify the main players etc...

[Figure]

From Bonnet, S., Caffin, M., Berthelot, H., and Moutin, T.: Hot spot of N2 fixation in the western tropical South Pacific pleads for a spatial decoupling between N2 fixation and denitrification, Proceedings of the National Academy of Sciences of the United States of America, 114, E2800-E2801, 10.1073/pnas.1619514114, 2017.

Figures and tables – Need more of them!, and more quantitative! ODV color contour plots are nice, but awfully hard to interpret quantitatively, and quite impossible if looking at a B/W version of your paper. Show some convincing/representative profiles, hard contour lines and East-West quantitative values. The discussion, by and large, is mostly hand-waving and speculation. While quite a few papers are cited (several of which haven't been published), there is a lack of quantitative information and data presented from these studies with which to compare the authors' results, assess their veracity and draw comparisons. The bit about regional differences being due to iron (etc.) inputs from submarine volcanos is wholly speculative on the information provided. Not a shred of quantitative information is presented to back this assertion up. Might the regional difference in fixation be due to regional differences in wind stress and water masses which affects the depth of the mixed layer and vertical mixing through the thermocline? I'd like to see a more focused discussion.

We now provide the vertical profiles of N2 fixation in the supplementary information. The vertical and longitudinal variability of N2 fixation are now better described in the results section and discussed in the section 4. The depth of the mixed layer has been also added to the revised manuscript and is homogeneous across the zonal transect and may not explain the differences observed (all this is now presented and discussed). The full set of dissolved iron data (see figure below) cannot be provided in the revised version as they are used in another paper under (minor) revision in Scientific reports. However, the discussion on the role of iron has been updated page 14 lines 9-34 to give more quantitative data.

[Figure]

Surface Chlorophyll-a concentration (mg m⁻³) during the 45-day transect of the OUTPACE cruise (A) (The ocean color satellite products are produced by CLS. Figure courtesy of A. De Verneil). (B) Cross-section of dissolved Fe nM (0-500 m). From Guieu et al., (Under review, minor revisions)

Why no comparison with the extensive work done on N-fixation, fluxes and driving processes done at station ALOHA? An opportunity is missed.
We performed an extensive comparison between the OUTPACE results and the ALOHA station data in our other OUTPACE paper by Caffin et al., (2018) regarding the functioning of the ecosystem, the role of N2 fixation on export etc…

Some specific comments:
Page Line(s) Comment 1 2 What exactly do you mean by 'hot spot' – see above 3
Please see comment above

10-11 By 'highest rates of N2 fixation' do you mean on an area-specific basis (I doubt it) or aggregate fixation on a regional basis primarily because of the very large area involved? The oxygen deficient zones of the eastern Pacific are due to higher regional productivity arising from Ekman-driven upwelling at the basin scale, not N-fixation. Indeed, fixation tends to be lower in upwelling regions. Simple N:P ratios are a poor predictor in this case.
Yes, I mean that Deutsch et al., (2007) predicted the highest rates of N2 fixation in the global ocean in the ETSP. I totally agree that this is a high productivity area mainly due to the upwelling, so it was counter intuitive to find N2 fixation there, but their argument was that the decrease of P* in this region would be the result of intense N2 fixation. This Deutsch et al study motivated many cruises in the ETSP to quantify N2 fixation there. Most of them found low rates (average range ~0-60 µmol N m⁻² d⁻¹, (Dekaezemacker et al., 2013; Fernandez et al., 2011; Fernandez et al., 2015; Knapp et al., 2016; Loescher et al., 2014)).

10-20 This paragraph seems to do a logical U-turn 21 etc. References to regional N-fixation ignore large database of published historical estimates by Capone, Carpenter, etc., etc., etc.

I am not sure to get this comment. In this paragraph we review the existing literature in the ETSP and WTSP and all along the manuscript we compare our results with the global MAREDAT database, which includes historical data from Capone, Carpenter... There are very few studies in this South Pacific. I wish I could cite more papers including Capone, Carpenter... I may have missed something, so if you know more papers published for this area, please provide the references. Some of them include Capone results especially the Knapp et al. 2016 and Dekaezemacker et al. for which D.G Capone is co-author (he was also the chief scientist onboard the cruise from which these papers originate). The Montoya et al., 2004 paper in the Arafura Sea also include D.G Capone data...

11 Change "equalled to" to "reliably extrapolated to estimates of"
This has been changed

Change "previously undocumented" to "new"
This has been changed

What is the basis of 'selection' – suggest leave out
nanoSIMS analyses are very expensive and time consuming. It was this not possible to perform such analyses at all stations, and we thus selected three stations to perform those analyses based on the diazotroph community composition assessed microscopically onboard. We remove the terms 'selected stations' in the introduction and specify in the Methods section that those analyses were performed at only 3 stations

What do you mean by 'potential ecological impact' of N-fixation? Poor wording.
We changed by 'biogeochemical impact'

Suggest changing 'contrasted' to something about a gradient of conditions. What's the essential difference between 'oligotrophic' and 'ultra-oligotrophic'? Strictly speaking that's like saying something is 'more better'.
The depth of the deep chlorophyll maximum was the main criteria. We changed the sentence as follows: 'It covered a trophic gradient from oligotrophy (deep chlorophyll maximum (DCM) located at ~80-100 m) in MA waters around New Caledonia, Vanuatu, Fiji up to Tonga, to ultra-oligotrophy (DCM located at 115-150 m) in GY waters located at the western boundary of the South Pacific Gyre (see Introduction article Moutin et al., 2017 for details on the cruise implementation)'.

(etc.) By 'fluorescence', I presume you mean 'chlorophyll fluorescence' –
say it because lots of other things fluoresce if you measure it right.
It has been changed

4 Use 'stored' instead of 'preserved. More importantly for low-level nutrient analyses –
how long were the samples stored before analysis (hours, days, months)?
This has been changed. The nutrient samples were stored for ~3 months before analysis (the time for the 4°C container to be back from French Polynesia (Tahiti) to Marseille (France). Previous experience from our team have confirmed that this way of conservation (HgCl2, 4°C) is a valid methodology.

7-10 Are there any actually published papers that describe these methods? Preferable. In the case of dissolved iron, the more widely used and less confusing notation would be: Fediss.
The manuscripts mentioned in the submitted version are now all available online on the webpage of the special issue https://www.biogeosciences.net/special_issue894.html and most of them have been accepted, except the Fe paper (presently in minor revision), so the method for dissolved Fe concentration determination is now given in this section.

In using open-ocean communities, it is almost universally observed that metal and organic contamination results in under-estimation of rates due to toxicity of these materials to finicky oceanic bugs. Why do you think they are over-estimated?

The most likely metal contamination onboard a metallic ship is iron, which would likely fertilize the studied water mass and consequently would over estimate rates (together with DOM contaminations) (see previous work from Bonnet et al., 2009). We acknowledge that other metals such as Hg or Cu at high concentrations would have the opposite effect due to toxicity, but this is unlikely to occur with the protocols we use as they are scarce compared to iron.

20-21 Presumably you mean sub-samples taken from the Niskins. Explicit reading suggests you collected the water in situ in the poly-carb bottles.

This was stated in the first paragraph of the method section (common to all other sub-sections of the methods) 'Seawater samples were collected by 12-L Niskin bottles mounted on the CTD rosette'. Wee re-specified that seawater was collected from niskin bottles here.

How were these filters stored and for how long? Text suggests they were analysed almost right-away (good if you can do it correctly!).

We stored the dried filters in a desiccator for 3-4 months before analysis. As long as there is no water anymore on the filters, they can be stored for months.

20 What's the "them"?

The polycarbonate bottles mentioned in the previous sentence. We changed as follows for more clarity: 'eight additional polycarbonate (2.3 L) bottles were collected from the surface (50 % light irradiance) to determine *Trichodesmium* and UCYN-B specific $N_2$ fixation rates by nanoSIMS and quantify their contribution to bulk $N_2$ fixation. Two of these bottles were amended with $^{15}N_2$ as described above for further...'

26-28 What are the flow cytometric characteristics you sorted and counted the UCYN cells with? A lot of those don't have much, if any photosynthetic pigment, and if they did, the near-surface ones would likely be bleached a bit? It would be nice to see a cytogram.

Pico- and nano- phytoplankton were clustered on FSC vs. SSC cytograms (left panel on the figure below) using 2 µm beads. UCYNs were associated to the nano- phytoplankton (right panel on the figure) and the UCYN associated cluster was established using orange fluorescence vs. red fluorescence cytograms. The cluster corresponding to UCYNs is shown in light green on the cytogram of the right panel on the figure, and could be clearly separated from other clusters.

[Figure]

2 Strictly speaking, you're 'estimating' this, not determining it.
Yes, this has been changed (etc) By 0-50 m, I presume you mean the 'surface mixed layer'. This is a key matter herein as the surface mixed layer thickness changes along your transect. Strictly speaking, surface is surface (say 0-5 m). Even small and ephemeral density gradients in the near-surface layer and surface mixed layer can have profound influences on vertical mixing rates and hence the light histories of cells embedded in the surface layer. Be very specific!
We changed the text to be more specific. 'The mixed layer depth (MLD) calculated according to the de Boyer-Montegut et al., (2004) method was located around 20-40 m throughout the zonal transect: Maximum temperatures were measured in the surface mixed layer (~0-20/40 m) and remained almost constant along the longitudinal transect with 29.1 ± 0.3°C in MA waters and 29.5 ± 0.4°C in GY waters'.
.

By DCM, I presume you mean 'deep chlorophyll maximum'?
Yes, it is now defined in the methods section.

29-34 It's not clear what, if anything, this paragraph contributes to the paper.
This paragraph is about a methodological aspect regarding the comparison of the solubilization of the 15N2 tracer according to the mode of incubation of samples used. This is the first time that such comparisons are done, and these results will be helpful for people who measure N2 fixation in on-deck incubators (a method widely used) and will hopefully convince them to perform MIMS measurements on their samples.

37-38 This seems very wrong. I'm presuming you actually mean the per-mil deviation of the particulate matter (δ15N) from the normal natural abundance of 15N (0.367%). Normally, N-fixation has a δ15N value close to 0‰ Larger deviations would suggest other fractionation processes such as denitrification. Clarify and fix up.
The term 15N/14N ratio was probably misleading. We changed the sentence by 'The natural N isotopic signature of suspended particles measured over the photic layer was on average -0.41‰ in MA waters and 8.06‰ in GY waters (Table 1)'.

1-8 This is all the 'new' N-fixation data in the results text. Pretty thin. As an interested party, I'd like to see a lot more. Graphics and tables too. 10-17 Correlations – So what?

Tables of correlation coefficients fill space, but are instantly forgettable. What's the point of the correlations other than that you can do them?

The presentation of N2 fixation rate results have been extended, we have merged sections 3.3 (N2 fixation results) and 3.4 (correlations) in the new version on the manuscript, and have added correlations between N2 fixation and primary and bacterial production as requested by another reviewer. Correlations are used in the discussion section and help the interpretation of the results.

18-23 Show 'em or ditch this.

Not sure to get this comment

1-5 etc Decimals on figures. It's easy to calculate lots of decimal places on figures, but they clutter up the text. Given the analytical and natural variability of these processes and the analytical processes, how many decimal places are really justified and meaningful?

For N2 fixation rates, given our quantification limit, one decimal only is justified and realistic. This is what is done in the manuscript.

17-18 Realistically, one never really 'measures' a process. Given all of the factors at work, the best we can do is 'estimate' its magnitude. Best to be frank about that.

We agree and replaced 'measurements' by 'estimates'

10-11 16-9 You probably overdid this bit of text. The 'bubble' problem is well known. Best to just say that you used the Montoya method to minimize contamination, but corrected for incomplete dissolution by measuring the 15N/14N ratio. It is interesting that you get higher dissolution in the samples incubated in situ and that needs to be explicitly corrected for

We shortened the text as suggested and removed the following part 'Two methods are routinely used by the scientific community to perform direct $N_2$ fixation measurements in marine systems: 1) the method developed by (Montoya et al., 1996), which consists of the addition of the $^{15}N_2$ tracer as a bubble in the incubation bottles (hereafter referred to as the 'bubble addition method') and the measurement of the $^{15}N/^{14}N$ ratio of PN before (time zero) and after incubation, 2) the method consisting of adding the $^{15}N_2$ as dissolved in a subset of seawater previously $N_2$ degassed (Mohr et al., 2010) (hereafter referred to as the '$^{15}N_2$-enriched seawater method'). The second method was developed because the first had been observed to potentially underestimate $N_2$ fixation rates (Großkopf et al., 2012; Mohr et al., 2010; Wilson et al., 2012) due to the incomplete (and gradually increasing during the incubation period) equilibration of the $^{15}N_2$ in the incubation bottles when injected as a bubble. This results in a lower $^{15}N/^{14}N$ ratio of the $N_2$ pool available for $N_2$ fixation (the term $A_{N2}$ used in the Montoya et al. (1996) equation) as compared to the theoretical value calculated based on gas constants, and therefore potentially leads to underestimated rates in some studies (see references above), whereas other studies do not see any significant differences between both methods (Bonnet et al., 2016c; Shiozaki et al., 2015)'.

Fig. 1 Bottom: longitude scale is very wrong.

The longitude scale provided here is the conventional one decided for all outpace papers. However we fixed that point.

Would like to see some comparative profiles of measured variables in different regions

The present paper is about the OUTPACE cruise results. We have added N2 fixation ranges from other part of the world for comparison, but this paper is not designed as a review of all the existing literature. Therefore we decided to stay focus on our data.

Fig. 2 ODV scales need to be properly annotated.

The x axis label has been added

Fig. 3 Potentially useful, but ... Bottom: should have x-axis scale 0-10 with vertical dotted lines clearly showing natural abundance of 15N (0.367%) and the theoretical 15N/14N ratio if all 15N2 in bubble dissolved. Is the (very) slight mid-water increase in 15N excess statistically valid?
This has been fixed